# Spatial multi-omics identifies aggressive prostate cancer signatures highlighting pro-inflammatory chemokine activity in the tumor microenvironment

Sebastian Krossa [1,2] ✉, Maria K. Andersen [1,3], Elise M. Sandholm[1,3], Maximilian Wess [1], Antti Kiviaho [4], Abhibhav Sharma [1,5], Sini Hakkola[4], Yangyang Hao[6], Mohammed Alshalalfa[6], Elai Davicioni[6], Trond Viset[7], Øystein Størkersen[7], R. Jeffrey Karnes[8], Daniel E. Spratt [9], Guro F. Giskeødegård [5], Matti Nykter [4,10], Morten B. Rye[3,11], Alfonso Urbanucci[4,12] & May-Britt Tessem [1,3] ✉

Understanding the characteristics of the tumor microenvironment (TME) associated with aggressive prostate cancer (PCa) is essential for accurate diagnosis and treatment. We interrogated spatially resolved multi-omics data to find molecular stratifiers of aggressive PCa. We report an aggressive prostate cancer (APC) gene expression signature predictive of increased risk of relapse and metastasis in a cohort of 1,588 patients. Further, we present a chemokine-enriched-gland (CEG) signature specific to non-cancerous prostatic glands from patients with aggressive cancer. The CEG signature is characterized by upregulated expression of pro-inflammatory chemokines, club-like cell enrichment, and immune cell infiltration of surrounding stroma. The activity of both signatures is correlated with reduced citrate and zinc levels and loss of normal prostate secretory gland functions. In summary we report that an increased inflammatory status linked to chemokine production, club-like cell enrichment, and metabolic changes in normal-appearing prostatic glands is associated with the subsequent development of aggressive PCa.

Prostate cancer (PCa) is the most common malignancy among men in western countries[1]. This disease is clinically, morphologically, and molecularly highly heterogeneous, which determines aggressiveness and clinical outcome[2]. Approximately 30% of PCa patients experience relapse after prostatectomy and improving patient stratification based on clinical and molecular parameters is an ongoing effort[3]. In search for biomarkers that are predictive of aggressiveness, substantial efforts are put into analyzing human PCa tissue having resulted in

[1]Department of Circulation and Medical Imaging, Norwegian University of Science and Technology, Trondheim, Norway. [2]Central Staff, St. Olavs Hospital HF, Trondheim, Norway. [3]Clinic of Surgery, St. Olavs Hospital, Trondheim University Hospital, Trondheim, Norway. [4]Prostate Cancer Research Center, Faculty of Medicine and Health Technology, Tampere University and TAYS Cancer Center, Tampere, Finland. [5]HUNT Center for Molecular and Clinical Epidemiology, Department of Public Health and Nursing, Norwegian University of Science and Technology (NTNU), Trondheim, Norway. [6]Veracyte, San Diego, CA, USA. [7]Department of Pathology, St. Olavs Hospital, Trondheim University Hospital, Trondheim, Norway. [8]Department of Urology, Mayo Clinic, Rochester, MN, USA. [9]UH Seidman Cancer Center, Case Western Reserve University, Cleveland, OH, USA. [10]Foundation for the Finnish Cancer Institute, Helsinki, Finland. [11]Department of Clinical and Molecular Medicine, Norwegian University of Science and Technology, Trondheim, Norway. [12]Department of Tumor Biology, Institute for Cancer Research, Oslo University Hospital, Oslo, Norway. ✉e-mail: sebastian.krossa@ntnu.no; may-britt.tessem@ntnu.no

tissue-based biomarker assays such as Decipher and Prolaris[4]. More recently, to capture the transcriptomic heterogeneity, single-cell and spatial transcriptomics (ST) have emerged as popular technologies to study PCa tissue samples[5–8]. ST has the advantage of capturing transcriptomics profiles in hundreds and thousands of locations across a tissue section thus offering a unique possibility to study the transcriptome in the spatial context, including intra-tumor heterogeneity, tumor microenvironment (TME) interactions, immune infiltration, and tissue-morphology-specific expression. Although single-cell transcriptomics data lacks spatial information, it has given new insight on cell-type specific transcriptome signatures present in the prostate[9,10]. Applying such derived detailed cell-type signatures to ST data shows considerable potential, as it has revealed highly detailed information on tumor-immune interaction in the prostate TME[6].

Inflammation is a risk factor in cancer including PCa[11,12]. Club cells can trigger inflammatory responses through chemokine secretion and are secretory cells initially characterized in the lung[13,14], but have been observed in the prostatic tissue[9]. In the healthy prostate, Club cells are located in the urothelial and proximal prostate zone, which was corroborated by studies in mice showing that club cells were present during early prostate development. In addition, they were proposed to contribute to benign hyperplasia by acting as multipotent progenitor cells[9,15]. Subsequent single-cell RNA-sequencing and ST studies observed club cell populations in the prostate peripheral zone in association with PCa[5,6]. These observations were also confirmed by the presence of Club cell markers in inflamed tissue adjacent to prostate tumor tissue[16]. The enrichment of Club cells in combination with reduced androgen receptor signaling in clinical tissue specimens has been shown as an effect of treatment with 5α reductase inhibitors and androgen deprivation therapy[17–19]. Through spatial transcriptomics analysis we recently demonstrated that Club-like cells are a key epithelial cell subtype associated with myeloid inflammation[20]. Inflammation and cancer lead to characteristic metabolic rewiring and the most prominent feature for PCa is the pronounced reduction of citrate levels compared to healthy prostate[21–23]. We have previously demonstrated that the diverse content of stroma can mask differences of metabolite and RNA levels between normal and cancerous PCa tissue in bulk analysis[24].

Here we used spatially resolved methodology integrating several -omics layers to gain an elevated understanding of PCa. We integrated and analyzed multi-omics data including spatial and bulk transcriptomics and metabolomics data connected to detailed histopathology collected from each prostate tissue sample. We dissected the biology of the TME from aggressive PCa patients and from patients remaining relapse-free 10 years post-surgery. We generated two gene expression signatures with elevated activity in patients with aggressive disease, associated with club-like cell enrichment, and PCa-characteristic transcriptional and metabolic changes. Our approach allowed also the identification of transcriptomic markers for highly localized metabolic aberrations in morphologically benign appearing glands in the TME of aggressive PCa, which highlights the substantial potential of integrating spatial-omics data to unravel the complexity and heterogeneity of cancer tissue.

## Results

To explore the gene expression in PCa tissue in its spatial, histology context we employed 10x Genomics' spatial transcriptomics (ST) assay to collect data in 19854 circular spots from 32 tissue samples from radical prostatectomies of 8 PCa patients (Fig. 1a). Of the eight patients, five experienced progression (aggressive disease patient group) with both persistency (n = 4, PSA > 0.1 ng/ml) and relapse (n = 1, PSA > 0.2 ng/ml) within 37 months, in addition 2 of these had confirmed metastasis after 90 and 22 months. Three patients remained relapse free, termed non-aggressive, during a follow-up of 10 years although they had similar clinicopathological features at diagnosis

**Table 1 | Clinical data for all patients included in this study**

| | Multi-layered spatial' omics (n = 8) | | Bulk transcriptomics & metabolomics (n = 37) | |
| --- | --- | --- | --- | --- |
| | Aggressive (n = 5) | Non-aggressive (n = 3) | Aggressive (n = 27) | Non-aggressive (n = 10) |
| ISUP grade | ISUP2, n = 1 | ISUP2, n = 2 | ISUP2, n = 7 | ISUP2, n = 8 |
| | ISUP3, n = 1 | ISUP3, n = 1 | ISUP3, n = 10 | ISUP3, n = 2 |
| | ISUP4, n = 1 | | ISUP4, n = 4 | |
| | ISUP5, n = 2 | | ISUP5, n = 6 | |
| Mean age and range at surgery | 59 (53–73) | 60.7 (56–63) | 62.4 (51–73) | 59.8 (55–66) |
| Mean pre-operative PSA | 23.7 | 14.0 | 18.7 | 10.8 |
| Clinical T-stage[a] | T2c, n = 1 | T2c, n = 2[a] | T2c, n = 9 | T2c, n = 9 |
| | T3a, n = 2 | | T3a, n = 6 | |
| | T3b, n = 2 | | T3b, n = 12 | |

The table is divided by patients of which tissue samples were used for multi-layered spatial' omics and bulk transcriptomics and metabolomics.
[a]For one patient T-stage information was not available.

(Table 1 and Supplementary Table 1). The 32 tissue samples were labeled according to sample location (cancer, normal adjacent, or normal sample location, Fig. 1a). Each ST spot was assigned a histopathology class, which included cancer with International Society for Urological Pathology (ISUP) grade group, non-cancerous glands (NCG), stroma, lymphocytes, lymphocyte enriched stroma and cancerous perineural invasion (PNI) obtained through hematoxylin-eosin(-saffron) (HE[S]) staining (Fig. 1a and Supplementary Fig. 1a, Methods). Each of the 32 samples was also subjected to multi-layered spatial analysis using immunohistochemistry (IHC) staining of bacterial antigens and mass spectrometry imaging (MSI) to obtain spatial metabolomics data. Further, these 32 samples together with an additional 142 samples from the remaining 29 of the 37 PCa patients were subjected to bulk transcriptomic and metabolomic analysis using RNA sequencing and high-resolution magic-angle spinning (HRMAS) NMR (Fig. 1a).

### Aggressive prostate cancer (APC) and chemokine enriched gland (CEG) signatures derived from spatial transcriptomics data

To investigate differences in the transcriptome between aggressive and non-aggressive disease patients, we first identified sets of differentially expressed genes for each of the histopathology spot classes individually in the ST data. The genes were selected based on highest normalized spatial expression variability and deregulation combined with gene ontology enrichment (GOE)-based evaluation of those genes (Methods, Supplementary Figs. 1a and 2). The GOE revealed inflammatory and immune-response-related processes as a major difference between these patient groups and evidence hinting at an important role of chemokine-mediated chemotaxis of neutrophils and macrophages (Supplementary Data 1). We thus decided to emphasize our focus on chemokines by manually including all chemokines detected in the ST data into our signature generation pipeline (see Methods and Supplementary Fig. 1b). We found a set of 26 genes which were able to distinguish aggressive from non-aggressive disease patients across all histopathology classes (Supplementary Fig. 3). We termed this set of genes the aggressive prostate cancer (APC) signature, which contained 18 genes with higher expression in patients with aggressive disease (relapse and persistent), and 8 genes with higher expression in non-aggressive disease patients (Fig. 1b). Most of the upregulated genes (14 of 18) in the APC signature were directly related to immune response

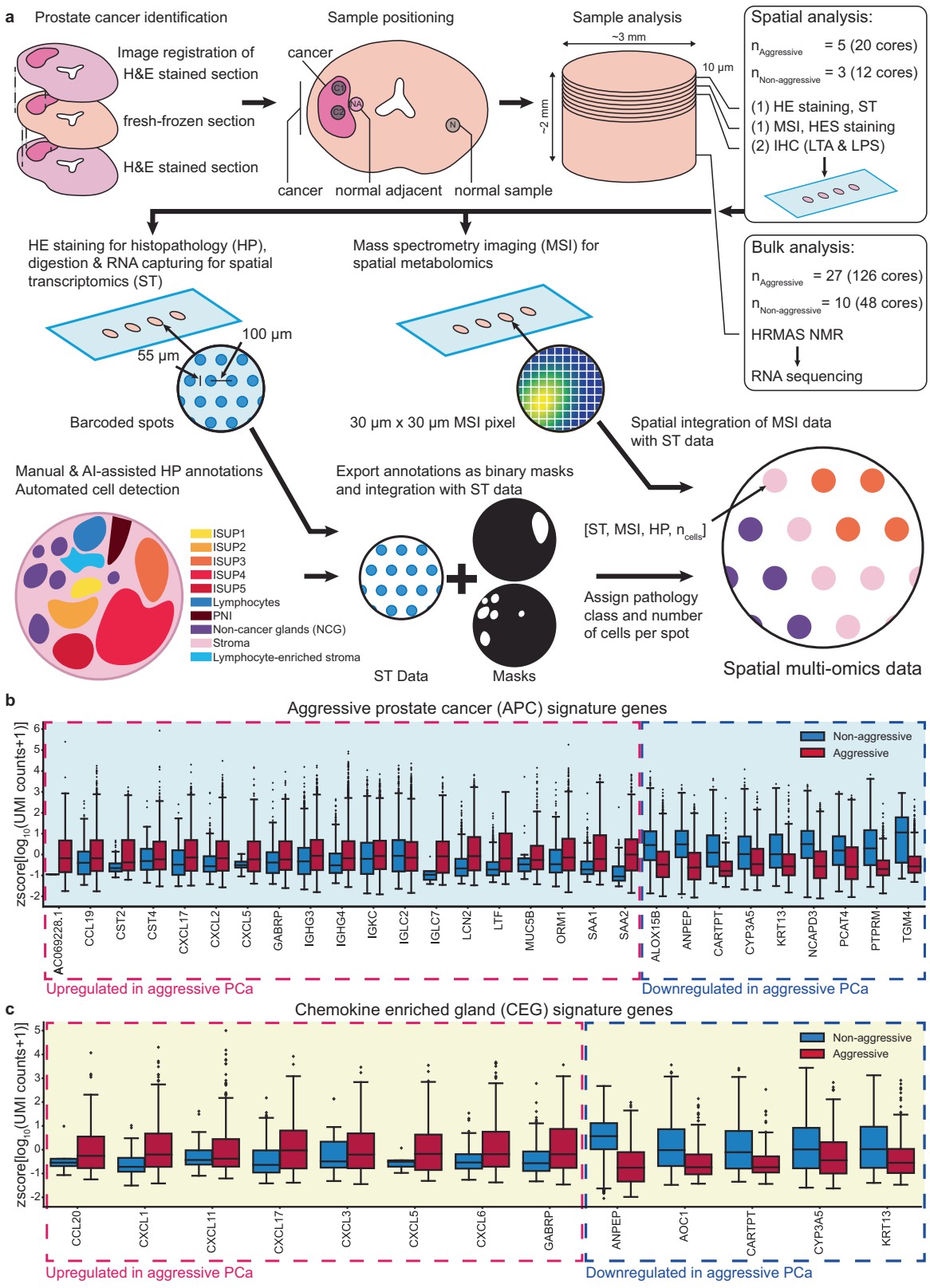

**b**  Aggressive prostate cancer (APC) signature genes

**c**  Chemokine enriched gland (CEG) signature genes

processes. In addition, we observed that the APC signature also scored high in some of the histologically benign NCG spots in samples both close and distant to cancer areas from patients with aggressive disease (Fig. 2a and Supplementary Figs. 1a and 3a). We therefore derived a gene set particularly capturing the transcriptomic changes in these potentially aberrant NCGs in the aggressive disease group. We termed this the Chemokine Enriched Gland (CEG) signature, reflecting the

observed upregulation of 7 chemokines in these benign appearing glands. We calculated for each spot the single sample Gene Set Enrichment Analysis (ssGSEA, Markert et al.[25]) scores for the APC and CEG signatures. The APC ssGSEA score showed an enrichment in all histopathology class spots and sample types from the aggressive disease group compared to non-aggressive disease patients (Fig. 2a, b and Supplementary Fig. 3a). Considering the distribution of the CEG

**Fig. 1 | Spatial multi-omics analysis of the prostate tumor microenvironment reveals gene signatures discriminating between aggressive and non-aggressive disease patient groups. a** Schematics of cancer, normal adjacent, and normal sample collection for bulk and spatial analysis of PCa tissue using bulk and spatial transcriptomics (ST), bulk metabolomics (high-resolution magic-angle spinning [HRMAS] NMR), spatial metabolomics (mass spectrometry imaging [MSI]), and histopathology staining (HE[S], immunohistochemistry [IHC] of lipopolysaccharide (LPS) and lipoteichoic acid (LTA)). The fresh frozen prostate tissue sections for sampling were selected based on pathology annotations from adjacent FFPE sections to locate cancer and normal areas. Cryo-sections from samples were obtained at 10 μm thickness and placed on slides for spatial analysis, here exemplary shown for ST and MSI. ST spots were classified according to per-section histopathology and assigned a cell count. Finally, all spatial modalities were integrated into one spatial multi-omics data set. Expression of up- and downregulated genes forming the **b** aggressive prostate cancer (APC) signature and **c** chemokine enriched gland (CEG) signature over all spots with a count >0 for the respective gene separated

according to status of aggressiveness (AC069228.1 $n = 1 + 1076$ [non-aggressive + aggressive], ALOX15B $n = 6195 + 6792$, ANPEP $n = 5674 + 5226$, AOC1 $n = 1530 + 668$, CARTPT $n = 775 + 305$, CCL19 $n = 449 + 3833$, CCL20 $n = 15 + 301$, CST2 $n = 369 + 3477$, CST4 $n = 116 + 421$, CXCL1 $n = 172 + 1373$, CXCL11 $n = 113 + 867$, CXCL17 $n = 349 + 1434$, CXCL2 $n = 328 + 1792$, CXCL3 $n = 58 + 654$, CXCL5 $n = 9 + 459$, CXCL6 $n = 193 + 669$, CYP3A5 $n = 1531 + 910$, GABRP $n = 197 + 971$, IGHG3 $n = 2371 + 9574$, IGHG4 $n = 1128 + 6009$, IGKC $n = 3269 + 10,999$, IGLC2 $n = 2598 + 9846$, IGLC7 $n = 6 + 1390$, KRT13 $n = 953 + 532$, LCN2 $n = 925 + 3808$, LTF $n = 970 + 5478$, MUC5B $n = 22 + 437$, NCAPD3 $n = 6155 + 6565$, ORM1 $n = 688 + 2336$, PCAT4 $n = 4234 + 2934$, PTPRM $n = 3529 + 2342$, SAA1 $n = 569 + 3544$, SAA2 $n = 105 + 1782$, TGM4 $n = 906 + 1766$). The box and whisker plots (box spans inter-quartile range (IQR), centerline indicates median, whisker extend 1.5 IQR from first and third quartile, observations beyond shown as individual points) are based on the z-score of the decadic logarithm of the normalized unique molecular identifier (UMI) counts.

signature, the ssGSEA score visualizes its capability to discriminate a sub-group of NCG spots. Although we observed an overall CEG signature enrichment in aggressive compared to non-aggressive disease patient groups, the spot histopathology class with the strongest separation were the NCG spots (Fig. 2c). Interestingly, not all NCG spots of the aggressive disease group had a high CEG score, some even had negative scores (compare Supplementary Figs. 1a and 3b NCG spots at 9 o'clock of section P30−sample N). In case of high CEG scores, the surrounding stroma was often infiltrated by immune cells as can be seen in sections P08−sample F & N, P30−sample C2 (compare Supplementary Figs. 1a and 3b). To further quantify the histopathological composition captured by both signatures we grouped the ST spots according to the APC and CEG ssGSEA score into groups with significantly high, low, or not significantly changed score (Supplementary Fig. 4). We found that for both signatures the not significant spot groups had similar compositions of approximately 40% stroma, 23% NCG, 35−40% cancer, and 1% lymphocyte spots. For the high APC score spots, we found a slight increase of stroma spots (56%) and a reduction of cancer spots (18%) while the amount of NCG spots remained stable (22%). The high CEG score spots were composed of an increased amount of NCG spots (55%) and a reduced amount of stroma (26%) and high grade (ISUP ≥ 3, PNI) cancer spots (3%). In contrast, the low score groups for both signatures consisted nearly completely of NCG spots (APC 80%, CEG 95%). In conclusion, we found two ST signatures with the APC signature displaying the aggressive transcriptomic changes independently of the histopathology while the CEG signature was characteristic for NCG and low-grade cancer spots of aggressive disease patients.

## APC and CEG signatures are associated with loss of luminal features and citrate secretion in glands

Considering that both our gene signatures contained a high number of immune-related genes, we investigated potential associations with molecular processes and cell-types. Further, based on the observation that both signatures highlighted a subset of high scoring NCG spots, we investigated if these spots, despite their benign appearing histology, already underwent molecular aberrations associated with cancer. Therefore, we calculated for each spot ssGSEA scores using 38 gene sets characteristic for cell types and cellular phenotypes related to either normal prostate or PCa tissue (Supplementary Fig. 5 and Supplementary Table 2). We suspected the distance to cancer could influence such changes and thus defined ten pseudo-bulk groups of ST spots based on the three major histology classes (cancer, NCG, and stroma), sample type (cancer, normal adjacent, and normal) and patient status (aggressive vs. non-aggressive disease) to simplify the analysis while preserving to some degree the spatial nature of the data. We merged cancer and normal adjacent samples unifying spots close to cancer and to obtain reasonable spot

numbers in each of the ten pseudo-bulk groups (863 to 4263 spots per group) to allow a reliable ssGSEA. We first validated our ssGSEA approach using gene sets for stroma and gland, the two most well-defined prostate tissue compartments[24]. All stroma spot groups showed strong enrichment for the four stroma-related signatures (Fig. 2d). The gland-related gene signatures Epithelia_GH, Luminal_GH, and Citrate (see Supplementary Table 2 for details) were generally enriched in normal sample NCG spot groups with higher scores in spots from non-aggressive disease patients (Fig. 2d), confirming the validity of our pseudo-bulk approach for ST data. Further, we observed that these gland signatures were depleted in NCG spots close to cancer in the aggressive disease group but not in non-aggressive disease patients. Interestingly, CEG and APC signatures followed an inverse trend, they were enriched in the aggressive disease group where the gland signatures were depleted and vice-versa (Fig. 2e). Additionally, we observed normal gland signature enrichment in cancer spots from the non-aggressive disease group, but depletion in the aggressive disease group. In conclusion, we found that the high activity of both signatures in benign appearing glands was associated with transcriptional loss of luminal features and citrate secretion.

## APC and CEG signature are associated with Club-like cell and immune cell enrichment

Considering the observed associations of the APC and CEG signatures with reduced luminal characteristics we wanted to gain a better understanding of underlying processes by looking at potential associations with specific cell populations present in the tissue of the aggressive disease group. Thus, we first investigated which of the cell-type specific gene sets of the 38 used in the pseudo-bulk ssGSEA approach were co-enriched in the same tissue regions, specifically in NCG spots close to the cancer (Supplementary Fig. 5b, CAg group). We noted an enrichment of immune cells from the lymphoid and myeloid lineages in this spot group (CAg) and even in the NCGs further away from the cancer (NAg) of the aggressive disease group compared to the respective groups (CIg & NIg) in the non-aggressive group (Supplementary Fig. 5b). Of note, we also observed an enrichment of the gene sets for Club and Hillock cells in these spot groups from the aggressive but not from non-aggressive disease group (Fig. 2e and Supplementary Fig. 5b).

Motivated by the correlation of our signatures with Club and Hillock cells, we further investigated signature and cell-type correlations on the individual spot level. Since ST data captures RNA from roughly 10−30 cells in each individual spot, the spot expression profiles originate from a mixture of cell-types. Thus, we choose to deconvolute each spot using *stereoscope*, a tool specifically developed to estimate cell-type compositions in ST spots by learning cell-type specific gene expression profiles from single-cell RNA-sequencing

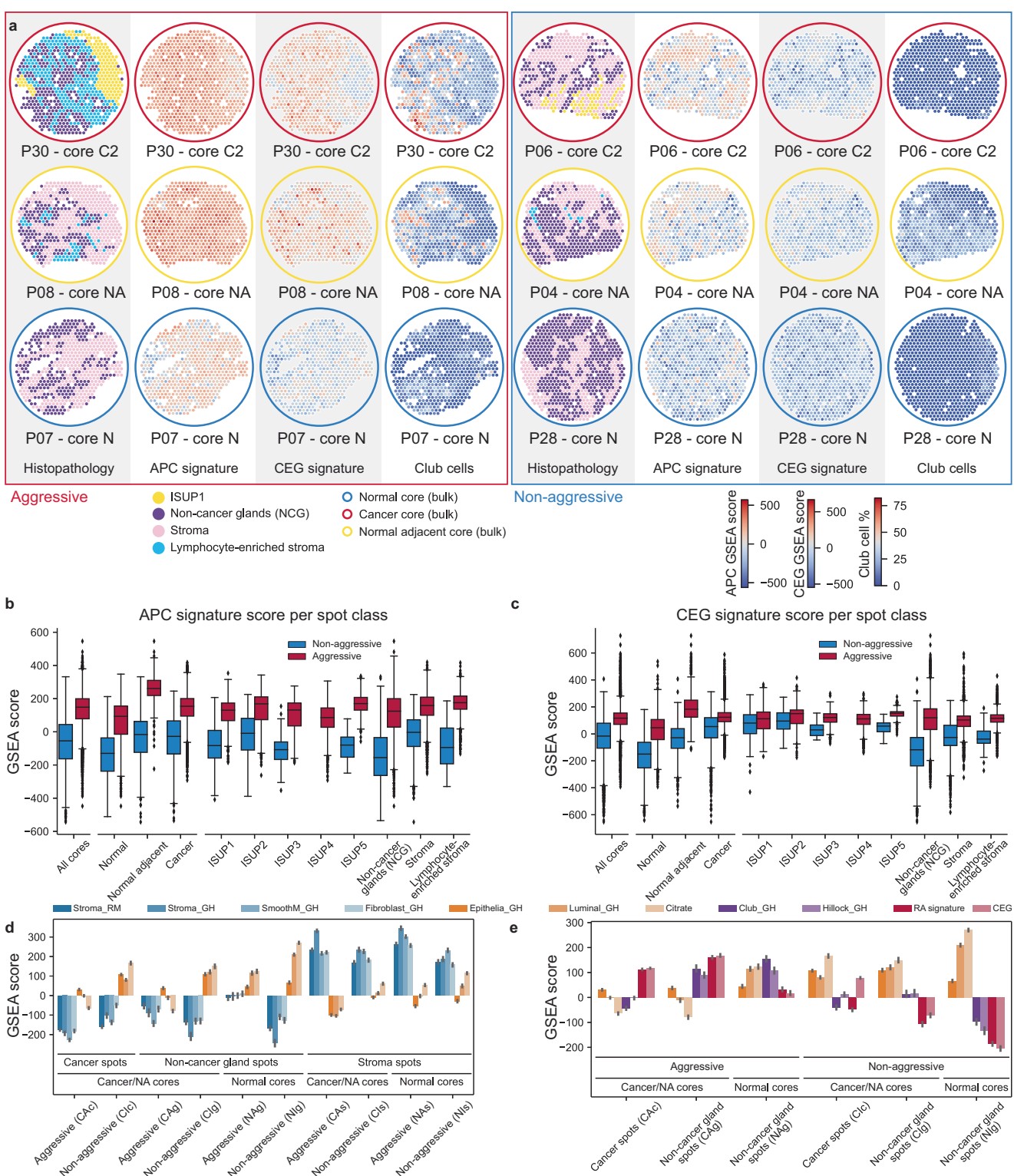

**Aggressive** / **Non-aggressive**

Legend:
- ISUP1 (yellow)
- Non-cancer glands (NCG) (dark purple)
- Stroma (pink)
- Lymphocyte-enriched stroma (cyan)
- Normal core (bulk) (blue outline)
- Cancer core (bulk) (red outline)
- Normal adjacent core (bulk) (yellow outline)

APC GSEA score: 500 / 0 / −500
CEG GSEA score: 500 / 0 / −500
Club cell %: 75 / 50 / 25 / 0

**b** APC signature score per spot class
(Non-aggressive / Aggressive)
GSEA score, categories: All cores, Normal, Normal adjacent, Cancer, ISUP1, ISUP2, ISUP3, ISUP4, ISUP5, Non-cancer glands (NCG), Stroma, Lymphocyte-enriched stroma

**c** CEG signature score per spot class
(Non-aggressive / Aggressive)
GSEA score, categories: All cores, Normal, Normal adjacent, Cancer, ISUP1, ISUP2, ISUP3, ISUP4, ISUP5, Non-cancer glands (NCG), Stroma, Lymphocyte-enriched stroma

**d** Stroma_RM, Stroma_GH, SmoothM_GH, Fibroblast_GH, Epithelia_GH
GSEA score; Cancer spots, Non-cancer gland spots, Stroma spots; Cancer/NA cores, Normal cores; Aggressive (CAc), Non-aggressive (CIc), Aggressive (CAg), Non-aggressive (CIg), Aggressive (NAg), Non-aggressive (NIg), Aggressive (CAs), Non-aggressive (CIs), Aggressive (NAs), Non-aggressive (NIs)

**e** Luminal_GH, Citrate, Club_GH, Hillock_GH, RA signature, CEG
GSEA score; Aggressive, Non-aggressive; Cancer/NA cores, Normal cores; Cancer spots (CAc), Non-cancer gland spots (CAg), Non-cancer gland spots (NAg), Cancer spots (CIc), Non-cancer gland spots (CIg), Non-cancer gland spots (NIg)

data[26]. The stereoscope model was trained on publicly available single-cell RNA-sequencing data obtained from normal prostate tissue[10].

Using this approach, we found that the overall cell-type distribution was dominated by epithelial cells, followed by a large fraction of smooth muscle cells, a comparably large fraction of immune cells, and smaller fractions of fibroblasts and endothelial cells (Fig. 3a). Interestingly, the most prominent difference between aggressive and non-aggressive spots groups were the relative increase of immune cells and Club-like cells (Fig. 3a). The correlation of histopathology spot classes with cell-type composition showed that NCG spots correlated

positively with epithelial cell types and negatively with fibroblasts and smooth muscle cells while stroma spots inversely correlated with these cell-types (Supplementary Fig. 6). Further, luminal epithelial cells were only weakly positively correlated with ISUP1 spots, and this correlation gradually decreased with increasing ISUP grade to reach a weak negative correlation with ISUP5 spots, reflecting the loss of luminal characteristics in this tissue. The correlation with epithelial cell types and histology are comparable to our observations using the ssGSEA pseudo-bulk approach and thus validated the underlying stereoscope model.

**Fig. 2 | APC and CEG signature activity in prostate tissue. a** Spatial distribution of APC and CEG signature activity, exemplary shown for one sample type for each patient group, visualizes the global and uniform activity of the APC signature as compared to the CEG signature. Spots with high CEG signature activity contained high estimated Club cell fractions. **b** APC signature activity was higher in all spot histology classes from aggressive disease patients (red) compared to non-aggressive (blue), while **c** CEG signature activity was higher in non-cancer gland (NCG) spots and less pronounced in stroma and cancer spots of aggressive disease patients and more prevalent in normal and normal adjacent samples. **b, c** ST data of 32 samples (non-aggressive $n = 12$, aggressive $n = 20$) from 8 (non-aggressive $n = 3$, aggressive $n = 5$) PCa patients shown as box and whisker plots (box spans inter-quartile range (IQR), centerline indicates median, whisker extend 1.5 IQR from first and third quartile, observations beyond shown as individual points; All cores $n = 7373 + 12,481$ [non-aggressive + aggressive spots], Normal $n = 1970 + 2499$, Normal adjacent $n = 1088 + 599$, Cancer $n = 4315 + 9383$, ISUP1 $n = 816 + 669$, ISUP2 $n = 1236 + 748$, ISUP3 $n = 81 + 484$, ISUP4 $n = 1452 + $ ISUP5 $n = 85 + 645$, Lymphocyte-enriched stroma $n = 96 + 2032$, Non-cancer glands $n = 2288 + 2724$, Stroma $n = 2771 + 3089$. **d** Mean ± SD scores of stromal activity gene sets (blue) in spots grouped by histopathology class (cancer, non-cancer gland, stroma), sample type (cancer & normal adjacent [NA], normal), and PCa aggressiveness. Stromal activity was increased in stroma spots and decreased in NCG spots and vice versa for selected epithelial gene sets (orange). **e** Mean ± SD activity of selected prostate epithelial gene sets, cell type specific gene set, APC, and CEG signature showed clear differences between aggressive and non-aggressive disease patients. Both CEG and APC signatures showed increased activity in aggressive disease patients accompanied by an increased activity of the Club signature and a slight reduction, especially in the NCG spots close to cancer, in luminal and citrate/spermine gene sets. **d, e** Data shown derived from ST data of 32 samples (non-aggressive $n = 12$, aggressive $n = 20$) from 8 (non-aggressive $n = 3$, aggressive $n = 5$) PCa patients, ST spots per group: CAc $n = 4272$, CAg $n = 1623$, CAs $n = 3754$, CIc $n = 2218$, CIg $n = 1181$, CIs $n = 2004$, NAg $n = 1101$, NAs $n = 1367$, NIg $n = 1107$, NIs $n = 863$.

To further narrow down cell type associations with the APC and CEG signatures, we calculated the Spearman correlation of the signatures with the stereoscope-derived cell fractions. We found that immune cells of the lymphoid and myeloid lineages, but not granulocytes, were positively associated with the APC and CEG signatures. Further, the strongest negative correlations were observed for APC and CEG signatures with luminal epithelial cells and the strongest positive correlations with Club-like cells (Fig. 3b, c). No clear correlation with hillock cells was observed (Fig. 3b). The correlation with Club-like cells agrees with our observation from the ssGSEA pseudo-bulk analysis, while for the hillock cells there were conflicting results. The *KRT13* gene is downregulated in both signature active regions (Fig. 1b) and hillock cells are characterized as KRT13+. Hence, the weak negative correlation of the stereoscope-derived hillock cell fractions with the signatures might reflect this. Nevertheless, Hillock cells are not well described in the literature, especially in the human prostate, making it likely that both approaches (stereoscope and ssGSEA) might capture other closely related cell types. Thus, we decided to focus on the enrichment of Club-like cells in NCG spots close to the cancer of aggressive disease patients that also had a depletion of luminal and citrate secretion signatures. Overall, loss of luminal characteristics and lower citrate secretion in these NCG spots indicated a switch from a healthy prostate gland phenotype to the molecular phenotype of PCa. In conclusion, these "defunctionalizing" NCG spots in the aggressive disease patient group were enriched with Club-like and immune cells while simultaneously characterized by high APC and CEG signature activity.

## APC and CEG activity correlate with presence of chemokine-secreting Club-like cells

To gain further insight into the cellular origin of the individual genes of our signatures, we looked at the correlation of those genes with the estimated cell type compositions (Supplementary Fig. 7). For the correlation of upregulated APC signature genes, we found Club-like cells to be among the highest positive correlated cell type for the genes *LCN2, LTF, SAA1, SAA2, CXCL17, GABRP, MUC5B, CXCL2*, and *CXCL5* ($\rho = 0.33–0.2$, Supplementary Fig. 7a). Grouping cells by lineage revealed that myeloid cells representing the first line of defense against infections were also the strongest positively correlated lineage with the same genes and *CCL19* ($\rho = 0.53–0.22$). The immunoglobulins *IGHG3, IGHG4, IGKC, IGLC2*, and *IGLC7* were highest positively correlated with lymphoid cells ($\rho = 0.39–0.18$). In contrast, luminal epithelial cells were the strongest positively correlated cell type with the downregulated genes *ALOX15B, CARTPT, NCAPD3, PCAT4*, and *PTPRM* ($\rho = 0.63–0.21$). For the CEG signature genes (Supplementary Fig. 7b), we found that for all the additional chemokines *CXCL1, -3, -6, -11*, and *CCL20* the strongest positively correlated cell type were the Club-like cells ($\rho = 0.2–0.1$) and on the lineage level, with the exception of *CXCL11*, the myeloid cells ($\rho = 0.19–0.1$). Of note, the expression of these genes, apart from *CXCL11*, was mostly negatively correlated with luminal epithelial cells and other cell types of the lymphoid or endothelial lineages. Considering that Club-like cells are known to secret chemokines[14], these correlations suggest Club-like cells as one likely source for the chemokines of our signatures.

## Atypical chemokine receptor 1 (ACKR1) is the dominant receptor for CEG-signature CXC-chemokines in the aggressive disease patient group

Chemokines interact with specific receptors typically resulting in chemotaxis of the target cells. After identifying Club-like cells as the likely source for the chemokine secretion captured in the CEG signature, we investigated which chemokine receptors were detected in the ST data and what potential receptor-ligand interactions were active. Of all the chemokine receptors, we only detected expression of *CXCR4* and *ACKR1* in the ST data (Supplementary Fig. 8a) of which only *ACKR1* is known to respond to the CXC chemokines found in the CEG signature. Our bulk transcriptomics cohort obtained from 174 samples of 37 patients (including the 8 patients with ST data) demonstrated that both *CXCR4* and *ACKR1* together with *ACKR3* and *CCR1* are amongst the highest expressed chemokine receptors in prostate tissue (Supplementary Fig. 8c). Nevertheless, to investigate potential signaling across different tissue types of the chemokines of the CEG signature in patients with aggressive disease, we performed a cellPhoneDB analysis of the ST data using the histopathology classes as groups. To avoid a potential bias caused by data sparsity, but also preserve some of the spatial information, we performed the analysis for all the aggressive disease group samples individually and combined the results but abstained from any further grouping using spot-to-spot distance or cell-type composition.

We found significant interaction of *CXCL1, -3, -5, -6*, and *-11* with *ACKR1* from NCG to lymphocyte, stroma, and lymphocyte-enriched stroma spots (Fig. 4a). Corresponding interactions were also observed from stroma and lymphocyte-enriched stroma to lymphocyte, stroma, and lymphocyte-enriched stroma spots. Interestingly, *CXCL1, 3*, and *6–ACKR1* interactions were also present with NCG spots as origin and ISUP1 and NCG spots as receiver. Of note, we found a *CXCL11-DPP4* interaction with NCG, stroma, and lymphocyte-enriched stroma spots as sender and NCG and cancer spots as receiver. The expression level of *CXCL1, -3, -5, -6, -11, ACKR1*, and *DPP4* in spots from the aggressive disease patients group supported these interactions (Fig. 4b). We found NCG and stroma spots to be the main source for these CXC chemokines while for the cancer spots the expression was lower and mainly detected in the low-grade spots. *ACKR1* was detected in all spot classes but was highest in both stroma classes and the lymphocyte spots. Compared to *ACKR1*, *DPP4* showed a high expression in all spot classes except for lymphocyte spots. To gain insight into the cell types that express *ACKR1* and *DPP4*, but also *CXCR4*, we looked at the

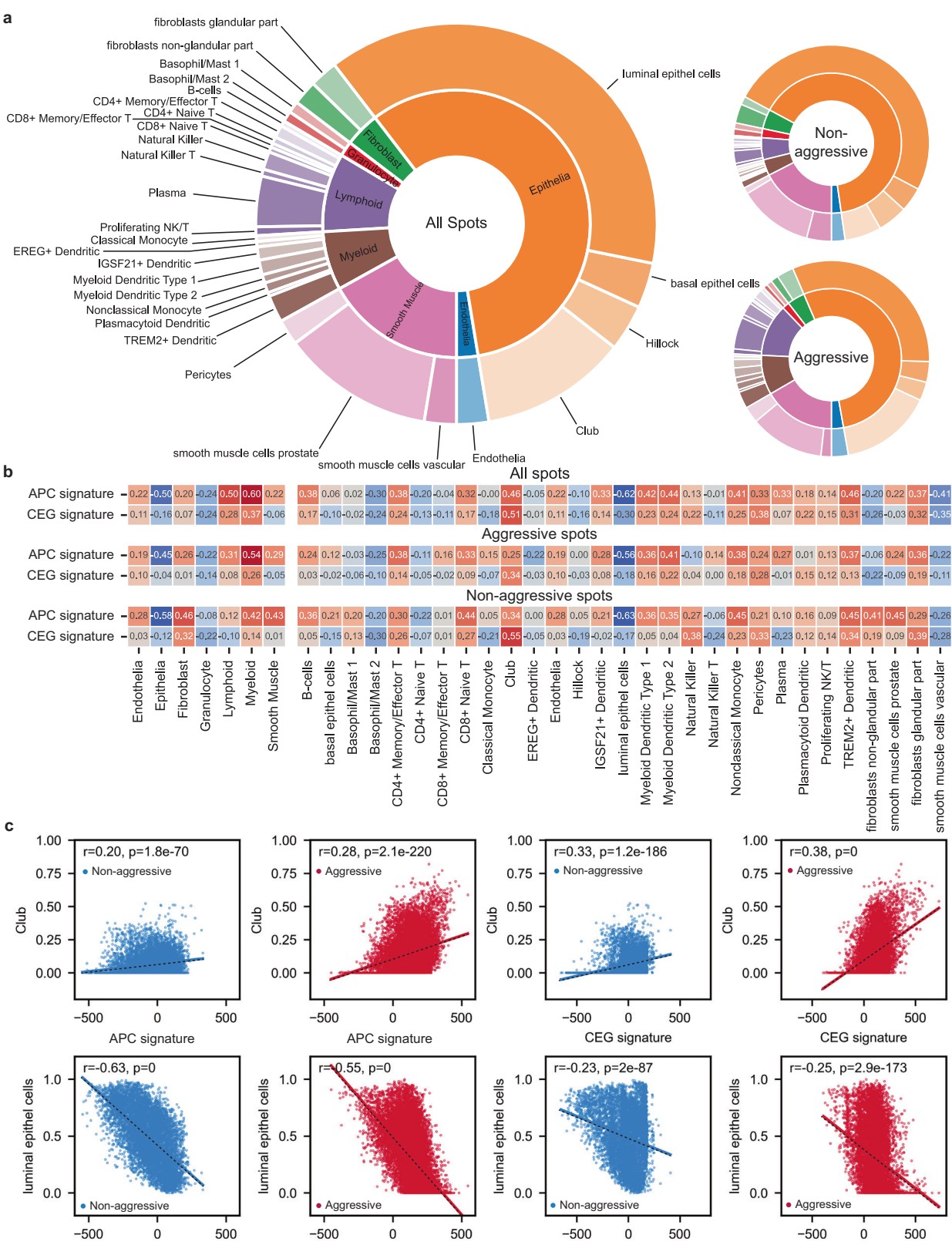

**Fig. 3 | Cell-type composition of each ST spot and correlation with APC and CEG signatures.** Using cell-type specific RNA expression profiles derived from single-cell RNA-sequencing data we were able to deconvolute each ST spot into its cell-type composition. **a** The accumulated cell-type fractions are shown for all spots (left, $n = 19,854$) and separated by status of aggressiveness (right, aggressive $n = 12,481$, non-aggressive $n = 7373$) of 32 samples (non-aggressive $n = 12$, aggressive $n = 20$) from 8 (non-aggressive $n = 3$, aggressive $n = 5$) PCa patients. **b** Spearman correlation coefficients calculated for cell-types and signature activity for all spots (top), spots

from the aggressive disease group (middle) and spots from the non-aggressive group (bottom) visualized by color (red = positive, blue = negative correlation).
**c** Scatterplots with linear regression of APC and CEG signature ssGSEA score vs. Club and luminal epithelial cells contents individually for spots from the aggressive (red, $n = 12,481$) and non-aggressive (blue, $n = 7373$) disease group. The R- and p values (Wald test with t-distribution, two-sided) for each linear regression are given inside the box of each plot.

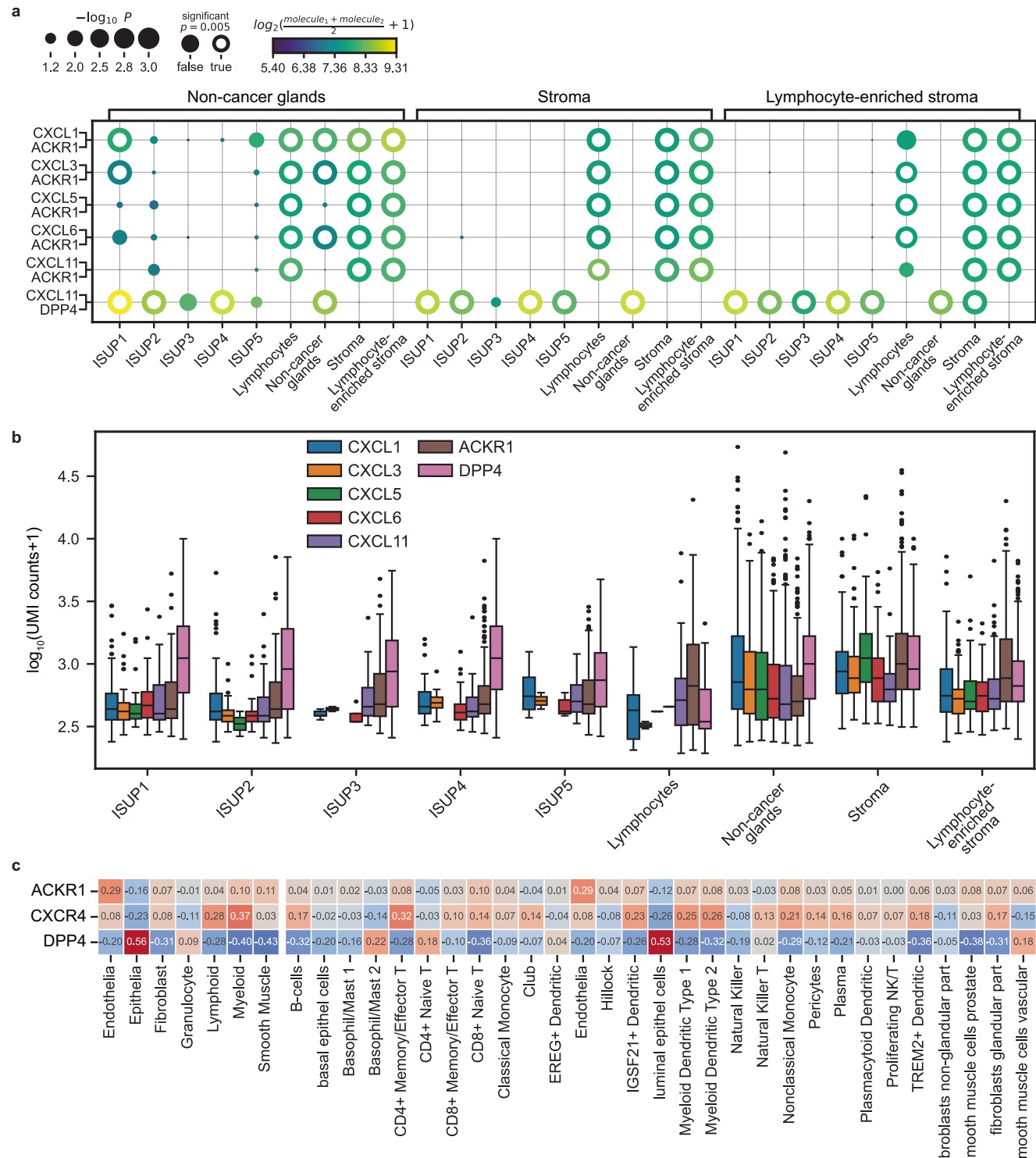

**Fig. 4 | Chemokine receptor-ligand interaction in tissue from patients with aggressive PCa. a** Potential receptor-ligand interactions of the CXC chemokines of the CEG signature with *ACKR1* and *DPP4* between different tissue types found by combining the CellPhoneDB results obtained from ST data of each of the samples from patients in the aggressive disease group (patients *n* = 5, samples *n* = 20, spots *n* = 12,481). The means of the average expression of the receptor-ligand pair are visualized with a color gradient and the *p* values (obtained by using CellPhoneDB's one-sided permutation test [*n* = 10,000] combined using Fisher's method, no multiple testing adjustment, chosen significance level *p* = 0.005) by the size of the circles. Receptor-ligand pairs are shown on the *y*-axis and interacting spot class pairs are given on the top and bottom *x*-axis. **b** Expression of the CXC chemokines of the CEG signature, the chemokine receptor *ACKR1*, and *DPP4* in spots with UMI count >0 from aggressive disease patient tissue shown as box and whisker plots

(box spans interquartile range (IQR), centerline indicates median, whisker extend 1.5 IQR from first and third quartile, observations beyond shown as individual points; number of spots: ISUP1 *n* = 127, 39, 27, 36, 59, 136, 544 [CXCL1, CXCL3, CXCL5, CXCL6, CXCL11, ACKR1, DPP4], ISUP2 *n* = 84, 28, 2, 28, 66, 159, 449, ISUP3 *n* = 5, 2, 0, 5, 21, 96, 177, ISUP4 *n* = 44, 12, 0, 22, 39, 200, 1077, ISUP5 *n* = 4, 2, 0, 7, 20, 95, 403, Lymphocytes *n* = 15, 2, 1, 1, 47, 94, 49, Non-cancer glands *n* = 701, 391, 266, 354, 388, 478, 1780, Stroma *n* = 224, 111, 94, 105, 62, 716, 778, Lymphocyte-enriched stroma *n* = 160, 65, 68, 104, 122, 474, 545). **c** Correlation of *ACKR1*, *CXCR4*, and *DPP4* expression with cell type fractions per spot using ST data from all 32 samples (non-aggressive *n* = 12, aggressive *n* = 20) from 8 (non-aggressive *n* = 3, aggressive *n* = 5) PCa patients, colored according to Spearman correlation coefficient (blue = negative, red = positive correlation). Correlations for the CXC-chemokines are visualized in Supplementary Fig. 6.

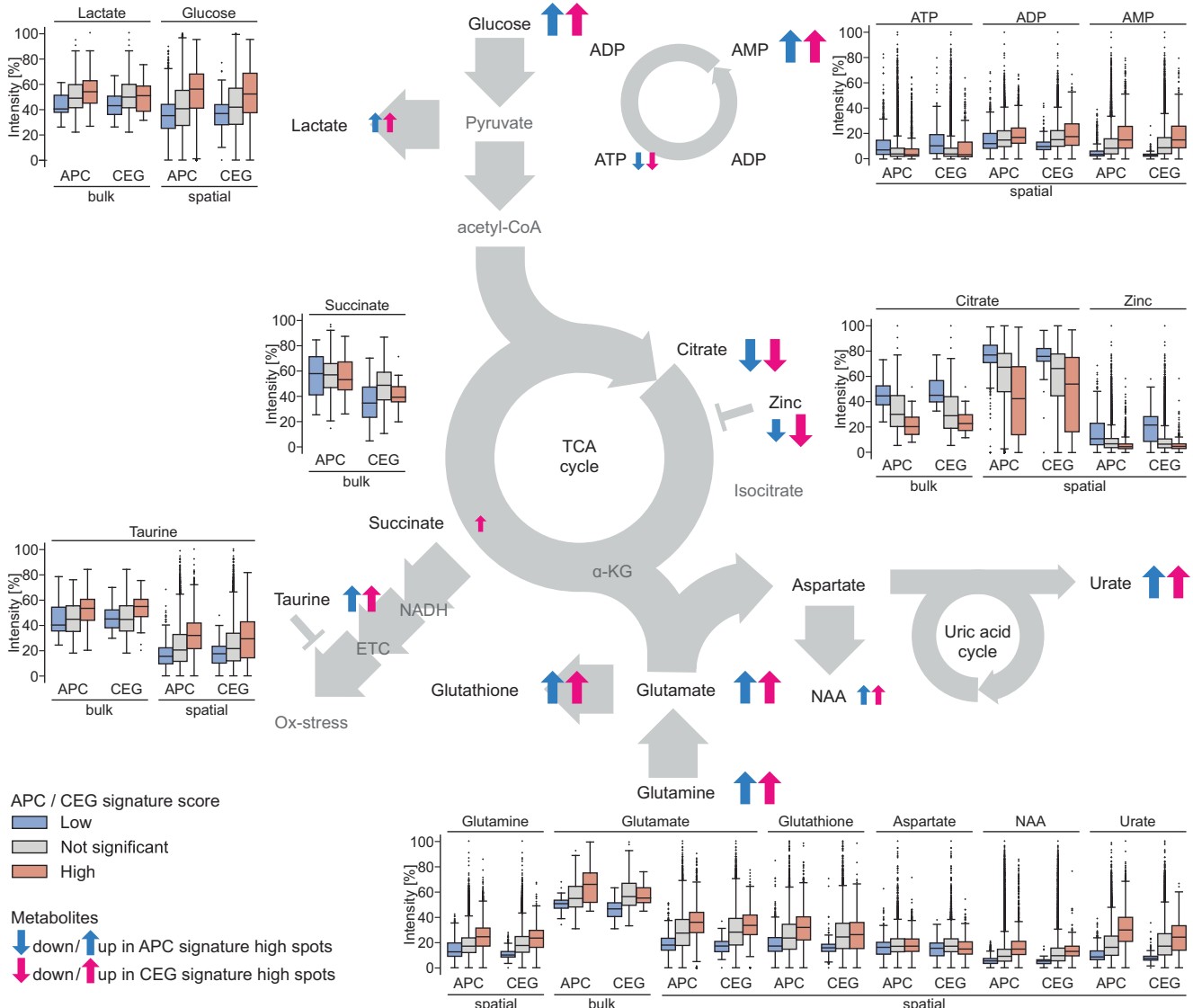

**Fig. 5 | Changes of metabolite levels in relation to APC and CEG signature activity.** Shown are the relative intensities in % as box and whisker plots (box spans interquartile range (IQR), centerline indicates median, whisker extend 1.5 IQR from first and third quartile, observations beyond shown as individual points grouped by signature activity of metabolites detected either by MSI (spatial, patients $n = 8$, samples $n = 32$, spots by class APC high, not significant, low $n = 2486$, 16,565, 731, CEG high, not significant, low $n = 603$, 18,984, 195) or HRMAS NMR (bulk, patients $n = 37$, samples $n = 174$, by class APC high, not significant, low $n = 35$, 114, 25, CEG high, not significant, low $n = 19$, 136, 19) placed according to their position in the indicated simplified metabolic pathways (glycolysis, TCA cycle, uric acid cycle).

Metabolite level changes (up or down) in spots (spatial) or samples (bulk) with an increased signature activity compared to low signature activity are indicated with up or down arrows (blue = APC, magenta = CEG signature). One-sided $p$ values obtained using permutation test ($n = 1000$) were used to assign significant high/low ST spots ($p$ value ≤0.05/number of signature genes [$p_{APC}$ ≤0.001, $p_{CEG}$ ≤0.003], absolute score >200, no further multiple testing adjustment) and bulk samples ($p$ value ≤0.05, absolute score >200, no multiple testing adjustment). Exact $p$ values for each spot/sample provided in source data tables for Supplementary Fig. 3 (ST), 10, and 11 (bulk).

correlation between their expression and the cell type composition for each ST spot (Fig. 4c). The strongest and single distinct positive correlation of *ACKR1* was observed with endothelial cells ($\rho = 0.29$) and the strongest negative correlation with luminal epithelial cells ($\rho = -0.12$). In contrast, *DPP4* was negatively ($\rho = -0.20$) correlated with endothelial cells and strongest positive correlation was observed with luminal epithelial cells ($\rho = 0.53$). *CXCR4* displayed the highest positive correlation with cells of the myeloid ($\rho = 0.37$) and lymphoid ($\rho = 0.28$) linages and, like *ACKR1*, also strongest negative correlation with luminal epithelial cells ($\rho = -0.26$). To confirm the cell type-specific expression pattern of *ACKR1*, we analyzed scRNA-seq data from prostate tumors and castrate-resistant prostate cancer (CRPC) needle biopsies[27,28]. Our analysis confirmed that *ACKR1* is predominantly

expressed by endothelial cells (Supplementary Fig. 8b). Additionally, at the CRPC stage, a subset of club-like and basal cells also exhibited *ACKR1* expression. To conclude, we found *ACKR1* to be the dominant receptor for CXC chemokines in high CEG signature scoring spots and likely expressed by endothelial cells.

### Cancer-associated inflammation not correlated with bacterial infection

Acute and chronic bacterial prostatitis are common[29] and thus a likely cause for the inflammation we observed in our tissue samples. Despite this, we could not detect expression of cytokines typically induced by bacteria during acute infection, such as IL-6 and TNF-α in the ST data. We further investigated this by staining serial sections for

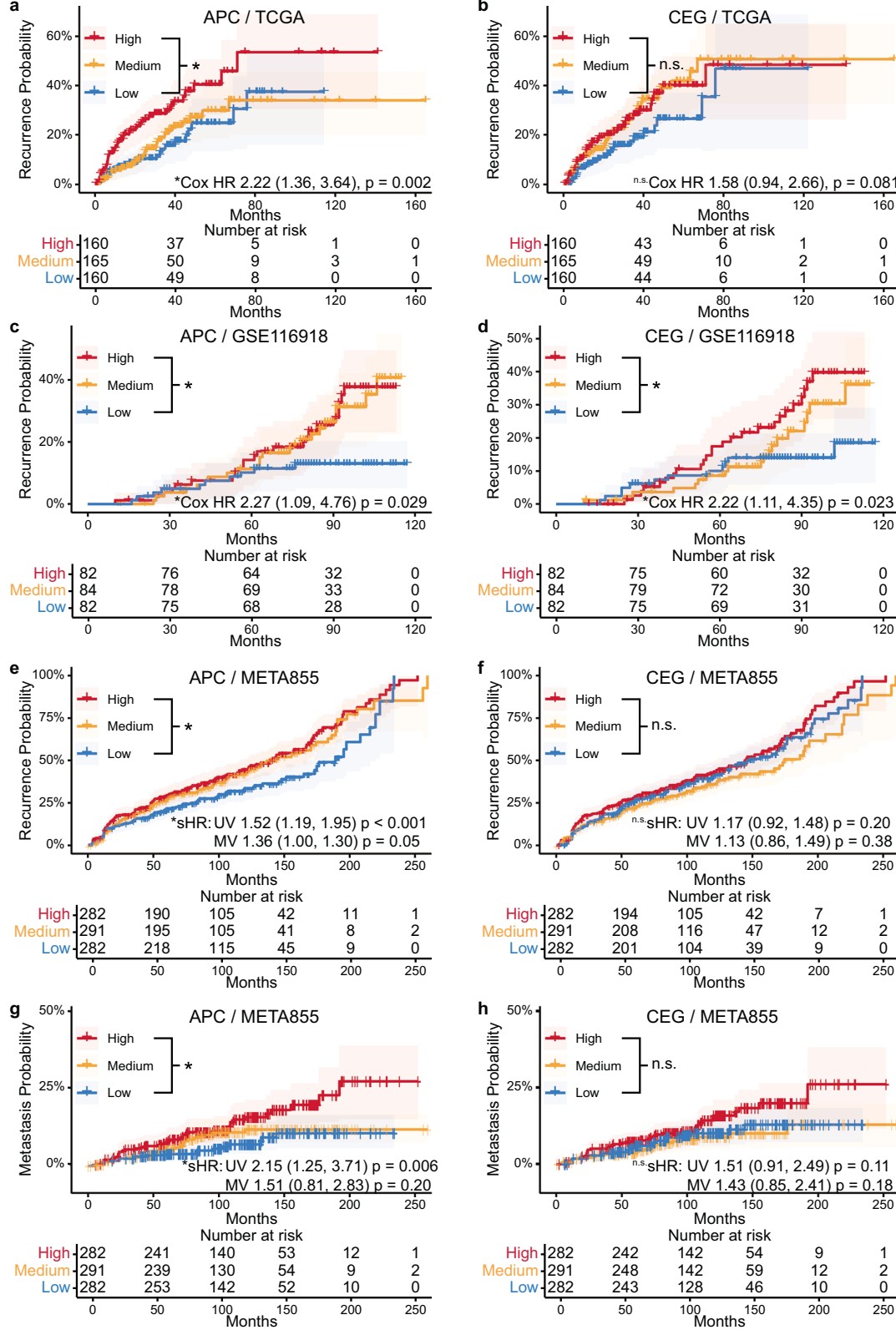

**Fig. 6 | Predictive power of APC and CEG signatures for biochemical recurrence and metastasis of PCa.** Shown are inverse Kaplan–Meier curves using APC (**a**, **c**, **e**, **g**) and CEG (**b**, **d**, **f**, **h**) signature activity for tertile grouping patients. Three external datasets were used TCGA (**a**, **b**, $n = 485$ cases), GSE116918 (**c**, **d**, $n = 248$ cases) and META855 (**e**–**h**, combined from the cohorts GSE72291, GSE79957, GSE62116 and GSE79915, $n = 855$ cases combined). Clinical endpoints are biochemical recurrence (**a**–**f**) and metastasis (**g**, **h**). Number at risk tables are given below each plot. Hazard ratios (HR) with their 95% confidence interval in parentheses are obtained from either a univariate Cox proportional hazards regression (Cox HR, **a**–**d**) or univariate (UV) and multivariate (MV) Fine-Gray subdistribution hazard model (sHR, **e**, **f**) and corresponding $p$ values obtained from the hazard models (univariate Cox: Wald test, two-sided; sHR: Gray's test, two-side, no multiple testing adjustment) are shown for the high vs. low tertile patients with the low tertile as reference. Significant HRs ($p \leq 0.05$) are indicated by an asterisk.

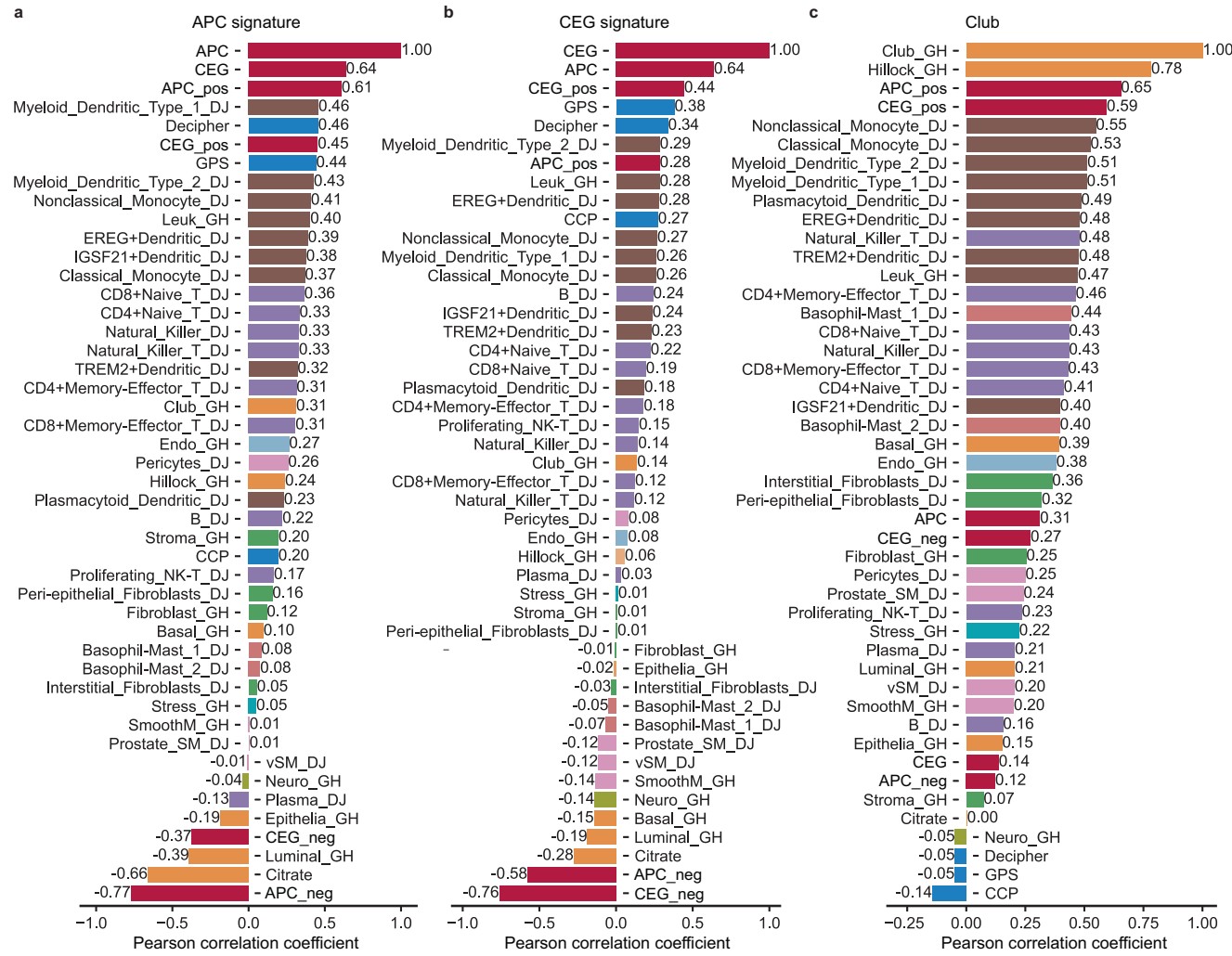

**Fig. 7 | Associations of APC and CEG with Club cell, other cell types, functional, and aggressive PCa gene sets.** Average Pearson correlation between gene sets across 12 public data sets ($n$ = 2688 samples in total) for **a** APC, **b** CEG and **c** Club cell signature. Correlations are displayed for APC and CEG signature with positive (upregulated genes) and negative (downregulated genes) counterparts (dark red), epithelial linage gene sets, like Club, Luminal cells, and Citrate gene sets (orange), immune cell gene sets (lymphoid−purple, myeloid−brown, granulocyte−light red), and genes from signatures known to be associated with aggressive PCa (GPS, Decipher and CCP, blue). Correlations for all gene sets and individual public data sets are shown in Supplementary Data 2.

lipopolysaccharides (LPS) and lipoteichoic acid (LTA) for the detection of gram-negative and gram-positive bacteria. Following affine co-registration of the serial sections we calculated an interpolated APC and CEG signature activity density map for each of these stained sections. Next, we ran an automated cell detection and assigned an estimated signature activity and staining intensity to each cell (Supplementary Fig. 9). Only a small fraction of the cells stained positive for LTA or LPS (359 of 718,901 and 374 of 746,976 cells, respectively) and we did not find an association of APC or CEG signature activity with LTA or LPS staining suggesting that it is unlikely that an active infection is the cause of the observed inflammation in glands found close to and within cancerous tissue.

## APC and CEG signature activity increased in bulk samples from patients with aggressive disease

The highly detailed, spatially resolved data from eight patients allowed us to find both signatures and characterize associated biological processes. Unfortunately, in day-to-day clinical practice such data is rarely available and such detailed analysis hardly feasible. However, prostate biopsies are typically collected and easier available for bulk molecular diagnostics. Thus, we investigated if the APC signature and the CEG signature were detectable and predictive of aggressive disease in a

larger PCa patient cohort where prostatic biopsies were gene expression profiled with a bulk RNAseq method. We collected and analyzed RNA-sequencing data from 174 PCa patients including non-aggressive disease ($n$ = 48), and aggressive disease ($n$ = 126) and from tissue samples classified either as cancer, normal adjacent, or normal, derived from 37 patients including non-aggressive disease patients ($n$ = 10) and aggressive disease patients ($n$ = 27) (Supplementary Figs. 10 and 11). The ssGSEA scores for the APC signature were higher in samples from the aggressive disease group (Supplementary Fig. 10a). The strongest signal originated from cancer samples while the difference in normal and normal adjacent samples was less prominent. Nevertheless, receiver operating characteristics (ROC) calculated using the median signature score per patient resulted in an area under the curve (AUC) of 0.75 (Supplementary Fig. 10b). A per patient analysis revealed strong patient to patient variations with at least one clear outlier in the non-aggressive disease group (P05, Supplementary Fig. 10c). Further, for the CEG signature we detected a weaker difference in activity between the aggressive and non-aggressive disease group as compared to the APC signature with an AUC of 0.66 of the ROC (Supplementary Fig. 11). Both groups had samples with high and low signature activity to the point that on average the signature activity was even slightly higher in normal and normal adjacent samples of

non-aggressive disease patients. A closer look at the per patient signature activity distribution (Supplementary Fig. 11c) revealed that this effect was mainly caused by patient-to-patient variability. For example, the non-aggressive disease patients P05 and P32 had samples with high signature activity while aggressive disease patients P09 and P25 had low signature activity. Interestingly, the overall difference—the variation around the median activity—was also much smaller in bulk data as compared to ST data.

**APC and CEG signatures are associated with altered metabolism**
After extensive and in-depth analysis of spatial and bulk transcriptomics data, we investigated if the loss of luminal features and citrate secretion on the transcriptional level in APC and especially CEG high scoring ST spots resulted in a measurable change of related metabolites. Further, we hypothesized that the inflammatory processes captured with our signatures were likely chronic and we investigated if they are correlated to metabolic aberrations. To reveal such changes, we examined spatial metabolomics data obtained by MALDI MSI on serial sections adjacent to sections that were used for ST (Fig. 1a). MSI data were registered to the corresponding ST data and median relative levels of putatively identified metabolites for each ST spot were calculated using our Multi-Omics Imaging Integration Toolset (MIIT)[30]. To analyze potential metabolic changes associated with APC and CEG signature activity, we applied the same grouping of spots into significantly high, low, or not significantly changed APC and CEG score as initially used for quantifying the histopathology composition captured by our signatures. Based on the composition of these groups, we considered the metabolite composition of the low signature score groups as most likely resembling healthy normal prostate glands while the not significant spot groups captured an average PCa tissue metabolic state. Consequently, we found that the signature low scoring spots had on average the highest levels of citrate and zinc while the not significant scoring spots had reduced levels of both (Fig. 5). For the high scoring spots this reduction was even more pronounced. Further, even though less pronounced, we found that ATP levels were also highest in low scoring spots as compared to not significant and high scoring spots. In contrast, we detected an inverse trend for AMP, ADP, glutathione, glucose, glutamate, glutamine, *N*-acetylaspartate (NAA), urate, and taurine. For all these metabolites we detected the lowest levels in low scoring spots and highest levels in high scoring spots. Aspartate was the only metabolite detected with similar levels in all spots. Interestingly, we detected the same trends and similar average relative metabolite levels for both APC and CEG signature high scoring spots despite their different histopathology compositions. Next, we analyzed the metabolite data obtained from our bulk samples by grouping, correspondingly to the spatially resolved data, by significantly high, low or not significantly changed signature activity (Fig. 5). We observed a comparable trend for citrate with the highest levels detected in low scoring samples and lowest detected in high scoring samples. Correspondingly, we found glutamate and taurine levels to be lowest in low scoring samples and highest in high scoring samples. Compared to spatial data these trends were less pronounced. Further, we found that lactate levels were slightly lower in low scoring samples as compared to the not significant and the high scoring samples. Interestingly, we found succinate to be the only metabolite with different levels between APC and CEG high scoring samples. Grouping bulk samples by APC signature score did not display any trend while for CEG, the low scoring samples showed comparably lower succinate levels. In conclusion, we found clear indications for metabolic alterations in high APC and/or CEG scoring spots in agreement with transcriptional loss of luminal functions and citrate secretions.

**APC signature is predictive for aggressive PCa in large public bulk data sets**
To test the clinically predictive power of APC and CEG signatures, we performed survival analysis in three publicly available cohorts and a cohort of multiple case-cohorts (META855, previously used to validate the Decipher prostate cancer genomic test) comprising a total of 1588 prostate cancer patients. The inverse Kaplan–Meier plots (Fig. 6) demonstrate that the APC signature is predictive for biochemical recurrence in all cohorts with significantly increased hazard ratios (HR) for patients with high APC scores. Further, using the META855 cohort, the APC signature was also predictive for metastasis with a significantly increased HR (Fig. 6g). Of note, using the meta data associated with the META855 cohort we obtained significantly increased HR using a multivariate Fine-Gray subdistribution hazard model for high APC scoring patients for biochemical recurrence and not for the multivariate but the univariate model for metastasis (Supplementary Tables 4–9). The HR for biochemical recurrence for patients scoring high in CEG was increased in all validation cohorts, however only significant for the GSE116918 cohort.

**APC and CEG signature are associated with Club-like cells and reduced luminal function**
To further substantiate biological associations, especially with Club-like cells, tied to our signatures, we analyzed bulk gene expression data from 12 publicly available PCa cohorts comprising a total of 2512 tissue samples (Supplementary Table 3). We calculated Pearson correlations between ssGSEA scores of APC, CEG, and Club gene set-based signatures with scores for the gene sets used previously in the ST data (Supplementary Fig. 5). As APC signature activity was predictive of aggressive PCa and none of the genes in our signatures overlap with the genes of three commercially used signatures (CCP, Decipher, and GPS) for aggressive PCa, we also included ssGSEA scores calculated using the genes from these signatures in this analysis. Correlation analysis in public data in general confirmed the associations observed in ST data (Fig. 7 and Supplementary File 3).

The APC signature correlated with genes from GPS ($\rho = 0.44$) and Decipher ($\rho = 0.46$), which supports the association between the APC signature and aggressive, recurrent cancer. The correlation was less strong for the CCP genes. The CEG signature also showed similar correlation preferences; however, the correlation values were weaker than for the APC signature. Both signatures also showed positive correlations to immune cells, but also here the correlation values were weaker for the CEG signature. Both signatures correlated negatively with the Citrate signature. Interestingly, the negative correlation to the Citrate signature was stronger compared to the negative correlation of Luminal cells, highlighting the relation of both our signatures with the secretory function of the luminal cells.

The APC and CEG signatures showed only intermediate (APC) or weak (CEG) positive correlation to Club-like cells in the public datasets. However, when we separated both signatures into their positive and negative contributions (up- and down-regulated genes respectively), we observed a strong correlation with Club-like cells for the positive parts of both signatures (Fig. 7c). This confirms that the observed associations between the two signatures with Club-like cells are extendable to larger patient cohorts. Interestingly, though inversely correlated in the ST data, the positive and negative counterparts of the CEG-signatures were not inversely but slightly positively correlated in the bulk data (Supplementary File 3). This could indicate that we are able to observe local compartmentalized signature combinations in ST data that is not detectable in bulk data.

## Discussion
Finding good biomarkers for PCa that can predict aggressive disease progression early is a challenging task and precise analysis of the relation of potential markers with known and potential new functional aspects of PCa is essential. Using ST data, we could identify two gene signatures, APC and CEG, of gene expression markers capturing changes in the TME of PCa and link them to biological functions and

metabolic changes. We validated the APC signature using large patient cohorts of bulk data including altogether 1588 PCa patients, where APC is predictive of PCa recurrence. The CEG signature was only slightly correlated and needs validation in future spatial data. The genes in APC and CEG do not overlap with clinically or commercially used signatures (such as CCP-Prolaris, GPS-Oncotype Dx, Decipher, or My Prostate 2.0-Lynx Dx) or recently published PCa biomarker panels[31–35].

Both signatures mainly captured upregulated inflammatory-related signals highlighting the role of an inflammatory immune response in aggressive PCa. Specifically, the APC signature (Fig. 1b) contained the acute-phase proteins serum amyloid A (*SAA1* and *SAA2*) and alpha-1-acid glycoprotein 1 (*ORM1*) as well as immune globulins (*IGHG3, IGHG4, IGKC, IGLC7*). The immune globulins can be seen as a direct gene expression-based indicator for the histologically observed increased presence of immune cells in tissue of the aggressive disease patient group. While serum amyloid A and alpha-1-acid glycoprotein 1 can be interpreted as markers for acute and chronic inflammation they have also been implicated in tumor pathogenesis and have been suggested as serum-based biomarkers for aggressive PCa and biochemical recurrence, respectively[36–38]. Cystatin SA (*CST2*) and Cystatin S (*CST4*) are known to be present in seminal fluid, have anti-viral properties and elevated levels have been reported for many cancer types including PCa[39–41]. Of note, we found lactotransferrin (*LTF*) upregulation to be associated with aggressive PCa in our data. While *LTF* has been reported as a tumor suppressor and immune modulator in PCa[42], a recent study in glioblastoma multiforme demonstrated that high *LTF* expression was associated with worse survival and increased immune cell infiltration[43]. Lipocalin-2 (*LCN2*) is known to be expressed and TNF-α-inducible in the prostate and a tumor and proliferation promoting factor[44,45]. Further, *LCN2* together with the chemokine CXCL1 were reported as prognostic markers for PCa relapse and facilitated metastasis in a mouse model[46]. The role of the mucin *MUC5B* in PCa has not been studied much but there are results implicating a role in hormonal escape of PCa[47]. Similarly, *GARBP*, the π-subunit of the GABA A receptor, has only been shown to be involved in in vitro PCa cell line proliferation[48]. Interestingly, our APC signature also contained the long non-coding RNAs *AC069228.1* and *PCAT4*. Their distinct role in PCa remains to be investigated, but two studies linked *AC069228.1* to relapse[49,50]. Reduction of *ALOX15B*, *PTPRM*, and *ANPEP* have been reported as progressors of and were associated with high-grade PCa[51–54]. Interestingly, *NCAPD3* has supposedly tumor-promoting effects in PCa[55] while expression was also associated with reduced recurrence after radical prostatectomy[56], thus confirming our observation that a reduction was associated with more aggressive PCa. Further, *CYP3A5* is known to be a modulator of androgen receptor signaling in PCa[57] while the role of *CARTPT*, a known oncogene, in PCa is not well understood[58]. In contrast, keratin 13 (*KRT13*) has been suggested as a marker for bone metastasizing PCa[59], while we found downregulation in PCa tissue to be associated with aggressive PCa. *KRT13* is also a hillock cell marker and appears to play a role in the development of the prostate but further research is required to uncover all functional aspects[60]. Downregulation of *SYCE1L* has not been reported as a marker of relapse or aggressive PCa before. The APC signature genes *LCN2, LTF, MUC5B, CXCL17* and CEG genes *CXCL1* and *CXCL6* are enriched in so called effector Club cells of the lung tissue playing a crucial role in the first line of immune response[14]. Our findings indicate that the observed enrichment of the chemokines *CXCL1, 3, 5, 6, 11, 17, CCL20* in the CEG signature was probably caused by a population of Club-like cells similar in function to lung effector Club cells in the respective NCGs causing the observed infiltration of immune cells potentially amplifying the chemokine signaling. Except for *CXCL17*, all of these chemokines are pro-inflammatory and attract immune cells[61] and are implicated in pro- or anti-tumor immunity depending on the respective chemokine ligand-receptor interaction and immune cell type[62].

Even though the APC signature was composed of many innate immune response factors and bacterial prostatitis has a high incidence[29], using IHC, we did not detect any signs of an active bacterial prostatitis as cause for the observed inflammatory response in the tissue. Regarding chronic bacterial prostatitis as a contributing factor, the results remain inconclusive. The detected low-level presence of LPS and LTA in regions with CEG signature activity might suggest some bacterial contribution possibly at an earlier time-point. The individual genes present in both the APC and CEG signature and their correlation with Club-like cells provide evidence of potential pro-inflammatory chemokine signaling from these prostate effector Club-like cells and a potential role in aggressive PCa. This shows further evidence on the important role of Club-like cells in immunological processes in the prostate that we recently uncovered[20]. The association of Club-like cells with our signatures was further supported by using independent publicly available data.

Our analysis of potential chemokine receptor-ligand interactions in the TME of PCa revealed *CXCR4* and *ACKR1* as the main receivers of CXC-signaling of which only *ACKR1* is known to bind the CXC-chemokines of our signatures. *ACKR1*, also known as the Duffy antigen receptor, was initially discovered on erythrocytes and loss of expression is associated with malaria resistance[63]. Apart from erythrocytes, *ACKR1* is expressed on endothelial cells of the lungs. Here *ACKR1* is involved in propagation of CXC chemokine signaling in inflammatory processes, for example by translocating chemokines from the basal to the apical side resulting in the recruitment of circulating immune cells. Considering that our data suggests *ACKR1* to be mainly expressed on endothelium in the prostate and the observed increased immune cell infiltration a similar role in the prostate as in the lungs appears plausible. Interestingly, *ACKR1*'s role in cancer has been reported as dominantly protective against tumor metastasis and uncontrolled proliferation, and is associated with positive outcomes[64]. Nevertheless, in PCa, reduced *ACRK1* expression on erythrocytes in men of African descent appeared not to be linked to aggressiveness[65,66] while *ACKR1*-deficient mice showed increased tumor growth[67]. In contrast, *ACKR3* expression together with *CXCR4* is associated with cancer promoting effects[68]. Even though, we only detected *ACKR1* and *CXCR4* in the ST data of those receptors, we found *ACKR3* to be equally high expressed as *ACKR1* in our bulk transcriptomics data. *DPP4* is known to cleave chemokines at the N-terminus resulting in altered signaling and diverse effects on inflammatory processes[69]. Considering that we found *ANPEP*, another peptidase known to correspondingly modulate *CXCL11* signaling[70], to be downregulated in the aggressive disease patient group, puts the proposed *CXCL11-DPP4* interaction into an interesting perspective strongly suggesting further studies on the role of peptidase-based alteration of chemokine signaling in PCa.

We investigated biological functions associated with our signatures by looking at the activity of known prostate function-related gene sets in our ST data (Fig. 2). Our results support our previously reported observations that prostate secretory genes are downregulated in tumors from relapse patients[52]. We could further substantiate this observation by analyzing metabolite compositions in APC and CEG active regions. Our findings show that high activity of both signatures is associated with metabolic changes in the respective regions. Specifically, the metabolic phenotype observed in APC high scoring spots displayed the typical hallmarks of PCa metabolism: Reduced citrate and zinc, increased levels of glucose, and signs for an increased glutamine consumption[71]. Interestingly, the high CEG signature regions, even though mainly containing NCG spots, showed a very similar, cancer-like, metabolic phenotype (Fig. 5). Zinc is typically understood to block aconitase to allow healthy prostate glands to accumulate high levels of citrate[72]. The observed reduction of both zinc and citrate, is a clear indicator for loss of this characteristic prostate gland function in NCG spots with high CEG signature activity. This agrees with our ST-based findings of reduced transcriptional

activity of citrate secretion associated genes in the same spots. This metabolic change of benign glands adjacent to cancer areas has, to our knowledge, not been previously observed in PCa. The loss of gene expression associated with citrate secretion in NCG spots from aggressive cancer samples (Fig. 2e) appeared to be the most profound change of the prostate specific luminal cell-type characteristics. When inspecting the genes present in the APC signature, we found that two of the downregulated APC signature genes (ALOX15B and NCAPD3) were also central genes in our previously published network of genes for citrate secretion[52], further strengthening the link between aggressive tumors and loss of citrate secretion in corresponding cancer adjacent benign glands. Further, using independent publicly available data, we could support the associations between both signatures and reduced luminal functions, specifically citrate secretion on the transcriptional level. The observed increased glucose levels in the spatial data in combination with the increased lactate levels in bulk data in signature high scoring spots and samples are indicators for anaerobic glycolysis also known as the Warburg effect[71]. Further, build-up of AMP and reduction of ATP, resulting in an elevated AMP:ATP ratio, are indicative for a low energy state of the tissue and suggest high activity of anabolic processes[73]. This was accompanied by a potential shift to utilization of glutamate as an energy source as an adaptation to a high energy demand under hypoxic conditions suggested by the pronounced increase of NAA, glutamate, and glutamine[71]. All these metabolic changes observed in regions with high APC and CEG activity are typically observed in cancer or inflammation[23,74]. The increased urate levels in these regions, indicative of a potentially increased activity of the uric acid cycle, might be a further indicator for a high energy demand. Interestingly, urate, and urate crystals, have been implicated in being a cause for prostatitis[75–77] and a factor leading to inflammation eventually contributing to prostate carcinogenesis[78]. The increase of taurine in these areas is interesting as it has been described as pro-apoptotic, anti-prolific, and anti-migratory[79] but also to mitigate the resulting oxidative stress from inflammation, typically increased in high catabolic tissue[80,81].

The bulk metabolomic data uncovered a comparable metabolic phenotype linked to increased signature activity as we found and discussed for the spatially resolved data. Both signatures are associated with loss of normal prostate glandular function indicated by a reduction of citrate and indicators for an increased catabolism. Bulk results in the context of the very localized CEG active NCG must be interpreted carefully. Nevertheless, the results from bulk metabolite analysis are in support of the spatially resolved data obtained using MSI for both signatures and in agreement with previous work as we discussed in the previous paragraph. Of note, the increase of lactate, a metabolite we could only detect using bulk HRMAS NMR, in samples with high CEG activity, might suggest that increased anaerobic glycolysis in CEG active NCG regions could lead to accumulation of lactate. High intracellular levels of lactate have recently been reported to result in chromatin remodeling and to negatively affect the response to anti-androgen treatment in prostate organoids[82].

This work suggests that early molecular changes happen in benign appearing glands of aggressive disease patients displaying high APC and CEG activity resulting in loss of luminal characteristics such as citrate secretion, and an aberrant metabolism typical for PCa. These results, mapping very localized features of the PCa TME, support previous findings that the switch to a molecular cancer phenotype happens early in PCa before morphological changes typical for cancer are unambiguously identifiable in the tissue[8,83,84].

The APC signature developed from spatial analysis was validated by using bulk transcriptomics data. Using ST data, we detected APC signature activity in all the major histopathology spot classes, including stroma and high-grade cancer, while the CEG signature was mainly active in some of the NCG regions in the TME. Thus, it appears reasonable that the predictive power of the APC signature is, to some

extent, independent of the spatial information while the CEG signature is precisely linked to tissue morphology. Further, despite our study design not consequently focusing on multi-foci sampling, the APC signature displayed relatively low intra-patient variability for most of the patients in our own multi-sample and partly multi-foci bulk data (Supplementary Fig. 10c), suggesting that using spatial data might alleviate the typically challenging heterogeneity in multifocal PCa for biomarker development[85].

Our results provide insights and convincing evidence to justify future research on larger cohorts using our spatial multi-omics approach and prospective studies to investigate the clinical applicability of the APC signature using bulk gene expression data.

In conclusion, using spatial multi-omics data obtained from the same samples, we generated two gene expression signatures capturing changes in the TME of PCa. Additionally, we were able to validate the ability of APC to predict aggressive PCa, higher risk of developing metastases and dying from PCa. In addition, high activity score of the generated signatures correlates with increased detection of Club-like cells, and transcriptional and metabolic changes, specifically changes of citrate secretion in normal looking glands of patients with aggressive PCa. Further, our data are suggestive of the fact that chemokines, major components in our signatures, are released by Club-like cells and are potentially responsible for the increased prostatic tissue inflammation. In conjunction with the loss of citrate levels and a metabolism shifted towards higher energy consumption, these findings suggest a critical role of chemokines in the development and progression of PCa by modulating the TME. Underlying the importance of focusing on targeting chemokines and their receptors as an attractive therapeutic strategy for combating PCa.

## Methods

### Ethical statement

The study utilized human prostate tissue samples obtained from PCa patients who gave informed written consent before undergoing radical prostatectomy at St. Olav's Hospital in Trondheim between 2008 and 2016. None of the patients had received any treatment prior to their surgery (further clinical details listed in Supplementary Table 1). This research received approval from the regional ethical committee of Central Norway (identifier 2017/576) and adhered to both national and EU ethical regulations.

### Statistics and reproducibility

In total, tissue samples from 37 patients were included in this study. Among them, ten patients, grouped as non-aggressive PCa, remained relapse-free for >10 years following surgery, while 27 patients, grouped as aggressive PCa, either relapsed ($n = 16$, biochemical recurrence, PSA > 0.2 ng/ml) or were persistent ($n = 11$, PSA > 0.1 ng/ml). In addition, two of the aggressive PCa patients had confirmed metastasis (details in Supplementary Table 1). We collected 174 samples (non-aggressive $n = 48$, aggressive $n = 126$) from all these 37 patients (non-aggressive $n = 10$, aggressive $n = 27$). Of these samples, we selected for spatially resolved methods 32 samples (non-aggressive $n = 12$, aggressive $n = 20$) from 8 of the 37 patients (non-aggressive $n = 3$, aggressive $n = 5$) choosing relapse patients with confirmed metastasis within 3 years. All samples were obtained by taking a 2 mm thick slice from the middle of the prostate (transverse plane) immediately after surgical removal. This slice was snap frozen and stored at −80 °C following the procedure described by Bertilsson et al.[86]. The collection of tissue slices and their storage were conducted by skilled personnel at Biobank1® (St. Olav's University Hospital in Trondheim). From each prostate slice we collected 8 to 13 tissue samples (3 mm in diameter) by using our in-house developed drilling device based on a previously described method[86]. For spatially resolved methods, we selected four samples per patient based on histopathology evaluation of hematoxylin-erythrosine-saffron (HES) stained tissue sections of each drilled sample. If applicable, we

acquired two samples with cancerous tissue, one sample with non-cancerous morphology close to the cancerous region, and one sample with non-cancerous morphology distant from the cancerous area for each patient (detailed images of sampling sites in Supplementary Data 3). No statistical method was used to predetermine sample size. Where applicable, details on statistical analysis are provided under each subsection. Exclusion criteria of individual data points of ST and MSI data are detailed in the respective subsections. Further, during IHC two LTA-stained sections had to be excluded because of partial sample loss during mounting. Exclusion criteria of samples from public data source TCGA-PRAD as detailed in the respective subsection. The order of data collection was randomized for all methods to limit potential technical batch effects. The Investigators were not blinded to allocation during experiments and outcome assessment.

### Tissue serial sectioning

For all tissue samples we collected several 10 μm-thick sections through serial cryosectioning at −20 °C using a Cryostar NX70 (Thermo Fisher). The tissue sections destined for spatial transcriptomics analysis were placed on Visium slides according to manufacturer's recommendations. Remaining tissue sections were collected for other methodologies (many of which not presented in this study) and were placed on other types of slides such as super frost, conductive slides and membrane slides. Sections destined for matrix-assisted laser desorption ionization (MALDI) mass spectrometry imaging (MSI) were placed on conductive IntelliSlides (Bruker Daltonics) and vacuum-packed as described by Andersen et al.[87]. All sections were kept cold during sectioning (−20 °C) and stored at −80 °C until further use. Material remaining after sectioning was stored at −80 °C until use for bulk metabolomics (HRMAS NMR) and transcriptomics (RNA-sequencing).

### RNA isolation, cDNA library generation and sequencing

For spatial transcriptomics profiling we used the Visium Spatial Gene Expression Slide & Reagent kit (10X Genomics) following the manufacturer's manual. Shortly before RNA extraction, tissue sections were fixated with 100% methanol, stained with hematoxylin and eosin (HE) and digitally scanned at ×20 magnification (Olympus VS120-S5). To ensure proper optical focus, a coverslip was temporarily placed on the tissue section and then gently removed after scanning. RNA was extracted by exposing the tissue sections to a permeabilization agent for 12 min. Optimal extraction time was previously determined with the Visium Spatial Tissue Optimization Slide & Reagent kit (10X Genomics). A second strand mix was added to create a complementary strand, and subsequently, cDNA was amplified using real-time qPCR. The amplified cDNA library was quantified using the QuantStudio™ 5 Real-Time PCR System (Thermo Fisher) through qPCR, and the cDNA libraries were stored at −20 °C until further use. Paired-end sequencing was conducted using the Illumina NextSeq 500 instrument (Illumina®, San Diego, USA) and the NextSeq 500/550 High Output kit v2.5 (150 cycles). Bulk RNA-sequencing of samples used in this study was performed as previously described[88].

### Generation of raw count tables

ST sample data were processed using the 10x genomics space ranger software package (version 1.0.0) according to manufacturer's recommendations. BCL files were demultiplexed to FASTQ files using spaceranger mkfastq. Raw count tables were generated from these FASTQ files together with the HE stained images and the human reference transcriptome GRCh38 version 3.0.0 per sample using spaceranger count. For the bulk samples, raw base call (BCL) files generated by Illumina sequencer were processed as previously described[88].

### Cell detection and number of cells per ST spot calculation

For each sample, cells were detected using the build-in-cell-detection feature of QuPath (version 0.2.3)[89] utilizing HE staining segmentation of nuclei to generate a list of nuclei centroids. For each spot, the number of centroids inside the spot area were counted to obtain the cells per spot count values. Data export from QuPath, merging with spatial transcriptomics spots, and cell counting were done using in-house developed groovy and python scripts and a python package. Code is available on github [https://github.com/sekro/qupath_scripts, https://github.com/sekro/spatial_transcriptomics_toolbox].

### Histopathology

Two experienced uropathologists (T.V. and Ø.S.) independently evaluated the HE-stained sections from all 32 spatial transcriptomics samples in QuPath (version 0.2.3). They annotated cancer areas according to the International Society for Urological Pathology (ISUP) Grade Group (GG) system[90] and aggregates of lymphocytes. Glands of uncertain cancer status were also annotated. We aimed to assign pure Gleason patterns to distinct cancer glands where possible. For example, well-defined separate areas of Gleason pattern 3 and 4 were annotated as IUSP GG 1 and 4, respectively, instead of one area of ISUP GG 2 or 3 (Gleason 3 + 4 or 4 + 3). For the downstream data analysis, a consensus pathology annotation was reached in agreement with both pathologists. Gleason scores were transformed into ISUP GGs and labeled with ISUP1-5 accordingly. Cancer areas with uncertain grading were annotated as ISUPX (indecisive between ISUP3 and ISUP5) and ISUPY (indecisive between ISUP1 and ISUP4). Other section annotations included tissue borders, lumen, glands, stroma, and stroma with higher levels of lymphocytes (referred to as 'lymphocyte-enriched stroma').

### Integration of histopathology with spatial transcriptomics data

Digital histopathology annotations including annotations for tissue bounds, luminal spaces, tissue folds, and uncertain areas were exported as binary images from QuPath using an in-house developed groovy script [https://github.com/sekro/qupath_scripts/blob/main/export_for_st_toolbox.groovy]. Histopathology data were merged with spatial transcriptomics data using an in-house developed python package [https://github.com/sekro/spatial_transcriptomics_toolbox]. For each spot, the intersection with the binary maps for each annotation class were calculated as a percentage per ST spot. The area of each of these intersections was divided by the tissue containing area of the spatial transcriptomics spot. The resulting percentage values per annotation class per ST spot were accumulated in a table and used to assign each spot a histopathology class. First, spots were excluded if they contained less than 50% tissue, more than 50% folded or low staining quality tissue, or more than 80% luminal space. If a spot contained more than 55% stroma it was assigned the stroma or lymphocyte-enriched stroma class using the one with the higher value. Otherwise, the spot was assigned one of the ISUP-grades, perineural invasion (PNI), or lymphocyte classes if the respective area of one of these classes was above 50%. In a last step, spots that were not assigned stroma or any of the ISUP-grades, PNI, or lymphocyte classes and contained at least 30% non-cancer glands (NCG), were assigned the class NCG. Spots not fulfilling any of the mentioned criteria were excluded from the analysis.

### Spatial transcriptomics data curation and normalization by number of cells per spot

Raw counts were normalized by dividing the counts by the number of cells for each spot (Fig. 1a and Supplementary Fig. 2). Normalized total counts per spot were calculated as the sum of all normalized counts and normalized number of detected genes per spot were obtained by counting genes with normalized counts >0 for each spot. Spots with

less than 100 normalized total counts or less than 40 normalized number of detected genes per spot were excluded.

Genes were only included if total raw count per gene over all spots was not zero and at least 10 counts for at least 10 spots were observed.

## Identification of APC and chemokine enriched gland (CEG) signatures

The ST spots were grouped according to the histopathology classes and analyzed for differences between aggressive and non-aggressive disease patients utilizing spatial information (see also Supplementary Fig. 1b). ST data raw UMI counts per cell values were decadic logarithm transformed prior scaling by 10,000 to obtain human readable values $(reads_{log 10} = log_{10}(reads_{\frac{raw}{n_{cells}}}*10000+1))$ and used for all downstream analysis described in this section. To capture the variable genes in the ST data, we used two approaches: (1) We calculated for each histopathology class the relative standard deviation (RSD, standard deviation/sample mean) for each gene and saved the 5 genes with the highest RSD per class into a preliminary gene list. (2) To find genes that are variable compared to genes of similar average expression, we used the approach suggested by Satija et al. and calculated for each class the dispersion (mean/variance) for each gene[91]. Then, genes were binned according to their mean into 20 bins. Subsequently, dispersions were standardized by their respective genes bin mean and standard deviation to obtain bin-wise "expression-level normalized" z-scores for each dispersion. We added the five genes with the highest z-score dispersion per class to the preliminary gene list. Next, we added deregulated genes in aggressive disease patients per class to the preliminary gene list. Deregulated genes were defined as genes with an absolute difference of the means of the normalized expression of aggressive and non-aggressive disease patients larger than 1 (corresponding to a ten-fold change in expression) and a maximum RSD of 20% in the non-aggressive disease patient group.

The preliminary gene list was filtered by removing genes with 75% or more observations coming from only one sample or patient and that have been observed in 2.5% or less of all spots. Further, to rank the remaining genes according to their strength to differentiate between aggressive and non-aggressive we calculated the min-max normalized zero-count-free median difference (ZFCMD) and the min-max normalized non-zero-counts spot count difference (NZCSCD), with:

$$ZFCMD_g = \left| median\left( c_{Aggressive,g} \right) - median\left( c_{Non-agressive,g} \right) \right|, \quad (1)$$
$$for\ c > 0, with\ c = normalized\ counts\ for\ gene\ g$$

$$NZCSCD_g = \frac{\left| n\left( S_{Aggressive,g,c>0} \right) - n\left( S_{Non-aggressive,g,c>0} \right) \right|}{\left( n\left( S_{Aggressive,g,c>0} \right) + n\left( S_{Non-aggressive,g,c>0} \right) \right)}, \quad (2)$$
$$with\ S = spots,\ g = gene,\ c = normalized\ counts\ for\ gene\ g$$

By excluding spots with 0 counts the ZFCMD can better capture intensity differences in the sparse ST data but gets more sensitive to outliers in cases of low number of spots with counts above zero. Thus, we calculated the NZCSCD to obtain a measure of the frequency of spots with counts above zero and derived a gene rank by calculating the average of ZFCMD and NZCSCD.

Genes with a rank-score below the median of the observed rank-scores were removed from the preliminary gene list. The filtered gene list was analyzed for enriched gene ontologies (GO) using the GOA-TOOLS python package[92] showing enrichment of immune and chemokine related GOs (chemokine activity, chemokine-mediated signaling pathway, antimicrobial humoral immune response mediated by antimicrobial peptide, inflammatory response). Thus, we added all chemokines observed in the ST data to the preliminary list and re-ran the filtering to obtain the final filtered gene list. The APC signature was derived from this list by accumulating all genes that were unanimously

up- or downregulated in all spot classes. The CEG signature was derived by accumulating genes that were exclusively up- or downregulated in aggressive NCG spots.

## Single-sample Gene Set Enrichment Analysis (ssGSEA) and score calculation

The ssGSEA approach is an in-house implementation of the algorithm presented by Markert et al.[25], which has also been used in a previous study by us on PCa samples[52]. To handle the sparsity challenge in ST data for ssGSEA, gene expression values were centered by subtracting the mean value for each gene before sorting. This will ensure that genes with zero expression in one sample will get a lower rank if it is highly expressed in other samples, compared to genes with zero or low expression across all samples. Scores were calculated separately for up- and downregulated parts of the APC and CEG signatures and combined by calculating the mean of the up- and negated downregulated ssGSEA scores. All ssGSEA scores were min-max scaled to a range between −1000 and 1000. Significant scores were determined by calculating a distribution of scores for $n = 1000$ unique permutations of random gene rankings. One-sided $p$ values for each sample/spot score were calculated from this distribution and combined for up- and downregulated part using Fisher's method. ST spot scores were considered significant when $p$ value ≤0.05/number of signature genes ($p_{APC}$ ≤0.001, $p_{CEG}$ ≤0.003) and an absolute score >200. For bulk samples a $p$ value ≤0.05 and an absolute score >200 was considered significant. Exact $p$ values for each spot/sample provided in source data tables for Supplementary Fig. 3 (ST), 10, and 11 (bulk).

## Subclassification in ten spot groups

Based on the classification of patients (aggressive/non-aggressive), samples (cancer/normal adjacent/normal) and spots (cancer, stroma, NCG) we assigned ST spots into ten groups. We combined cancer and normal adjacent samples into one cancer/normal adjacent sample class and combined all ISUP-graded spots into one cancer spot class. Spots classified as (lymphocyte-enriched stroma, PNI, and others) were excluded from these spot-groups. Consequently, the ten spot-groups were defined by having combinations of these classifiers: Level 1 (patient status): Aggressive, non-aggressive ($n = 2$), level 2 (sample class): Normal sample, cancer-normal adjacent sample ($n = 2$), level 3 (spot class): NCG, stroma ($n = 2$ for level 2: Normal sample) or NCG, stroma, cancer ($n = 3$ for level 2: Cancer-normal adjacent sample).

## Deconvolution of spatial transcriptomics data

Each spot of the spatial transcriptomics data was deconvolved into estimated cell type fractions using stereoscope[26]. For single cell reference data we used publicly available human prostate single-cell RNA-sequencing data (GSE172357)[10]. From this single-cell data we obtained a list of 4324 highly variable genes using Scanpy's implementation of the Seurat method[91,93]. Stereoscope was run with this list of highly variable genes and the following parameters: sc epochs = 75,000, sc batch size = 100, st epochs = 75,000, st batch size = 100, learning rate = 0.01.

## Receptor-ligand interaction analysis of chemokines of APC and CEG signature

To investigate what receptors are present and might act as receivers of the chemokines observed in the APC and CEG signatures squidpy's v1.2.3[94] implementation of the CellPhoneDB algorithm[94] was applied to the normalized and scaled but non-logarithm transformed (see section "Identification of APC and chemokine enriched gland (CEG) signatures") ST data. Omnipath[95] interactions were limited to "receptor" for the receiver and "ligand" transmitter and filtered to include only entries with at least one published reference. The histopathology classes assigned to the ST spots were used as cluster-key and permutation test was run with 10,000 permutations and threshold of 0.01 for

required cells per cluster. Values were calculated for each sample individually and corresponding interactions were subsequently combined by calculating the mean of the mean-values or combining the $p$ values using Fisher's method. Not-a-number (NaN) $p$ values were omitted during combination. In case all $p$ values for one interaction were NaN, we set the respective $p$ value in the combined-result tables to 1. Results were visualized with squipy's plotting function.

### Single-cell RNA-seq data processing for analysis of cell type-specific ACKR1 expression

The raw count matrices were analyzed with Scanpy[93]. Quality control of the scRNA-seq data from CRPC needle biopsies[28], downloaded from Gene Expression Omnibus (GEO) (accession no. GSE137829), was conducted by keeping cells with ≤9000 and ≥500 expressed genes and <10% of mitochondrial counts, and genes expressed in ≥15 cells. Additional quality control of scRNA-seq data from prostate tumors[27] (GEO accession no. GSE141445), was done to exclude genes expressed in <10 cells and cells with <600 expressed genes. Highly variable genes ($n = 2000$) were selected with Seurat v3 method for principal component analysis[96]. Normalization was done by total counts over all genes in each cell, then logarithmized and scaled, by centering each gene to have a mean of 0 and scaled to unit variance. Leiden clusters were annotated with cell type-specific marker genes.

### Immunohistochemistry staining of LPS and LTA

To estimate bacterial infection/presence in the 32 ST samples we performed immunohistochemistry staining of neighboring 10 μm cryo-sections of lipopolysaccharides (LPS) and lipoteichoic acid (LTA). Cryo-sectioned tissue was fixed in 4% formalin for 15 min, washed twice in TBS-T at room temperature. Subsequently, peroxidases were blocked using 0.3% hydrogen peroxide followed by blocking of unspecific binding sites using in 10% normal serum with 1% BSA in TBS for 2 h at room temperature. Primary antibodies against LPS and LTA (Abcam ab35654, clone 2D7/1, Lot No: GR3410325 and Thermo/Invitrogen MA1-7402, clone G43J) were used at 2.5 μg/ml and 2 μg/ml in TBS-T with 1% BSA and incubated on tissue over night at 4 °C. Anti-LPS and anti-LTA antibody dilutions were tested and validated against fixed and embedded E. coli and B. subtilis cultures. The tissue was washed twice in TBS-T prior incubation with secondary antibody (EnVision anti-mouse-HRP/DAB+ system, Agilent), 3,3′-Diaminobenzidine (DAB) development, counterstaining with hematoxylin, dehydration, and mounting were done according to manufacturer's recommendations. Stained tissues were scanned on an Olympus VS200 ASW 3.3 (Build 24382) slide scanner in bright field at ×40 magnification resulting in a resolution of 136,866 nm/pixel. Images were processed with QuPath using building functions for cell detection to obtain mean DAB stain intensity per cell. HE images of the ST data were registered onto corresponding DAB-stained images using affine transformation derived from 3 manually determined corresponding points for each image pair. For both APC and CEG signature, we calculated inverse-distance interpolated activity values for each cell center using the transformed ST-spot coordinates and signature activity as input (using k-dimensional tree, KDTree from python package SciPy, leafsize = 10, use 8 nearest data points, eps = 0.1).

### Metabolite detection with MALDI-MSI

Serial tissue sections from the same 32 samples that were used for ST were placed on conductive slides and analyzed with MALDI time-of-flight (TOF) MSI for the spatial detection of small metabolites and lipids. Prior to MSI, the sections were sprayed with matrix using the HTX TM-Sprayer™ system (HTX Technology). The matrix solution consisted of 7 mg/mL N-(1-naphthyl) ethylenediamine dihydrochloride (NEDC) in 70% methanol. Spraying parameters included 18 layers with 0.06 ml/min flow rate and 1200 mm/min nozzle velocity, 3 mm track spacing, 10 psi pressure, 2 l/min gas flow rate,

40 mm nozzle height, 75 °C nozzle spray temperature and 35 °C plate temperature. Following matrix application, MSI was performed with a rapifleX™ MALDI TissuetyperTM (Bruker Daltonics) equipped with a 10 kHz laser shooting 200 shots per pixel at a 10 kHz frequency. Negative ion mode was used with a mass range of $m/z$ 80–1000 and a spatial resolution of 30 μm. After MSI, the matrix-covered tissue sections were stored dry and dark at room temperature until HES staining was performed within 2 weeks. A selection of masses was identified as metabolites by MS/MS and accurate masses (<3 ppm) from our previous study using MALDI-Orbitrap[97], while Zinc was detected as an adduct with chloride ($ZnCl_3^-$) and was previously identified by its isotopic pattern[22]. All previous identification work was performed on fresh frozen PCa sections that had been collected and processed the same way as in this study and were covered by the same matrix. The identifications are therefore valid for this study and are on confidence level 2–3 as suggested by Baquer et al.[98]. Supplementary Data 4 lists all $m/z$ values and interval width and Supplementary Data 5 provides mean spectra and ion images of all metabolites used in this study.

### Integration of ST and MALDI-MSI data

MALDI-MSI data were binned/down sampled to 80% of its original datapoints in FlexImaging (Version 5.0, Bruker Daltonics) and further baseline corrected (top-hat) and root-mean square (RMS) normalized using SCILS lab Pro (Version 2024a, Bruker Daltonics). Ion images of identified metabolites (intensity given by max peak height) were exported to imzML files and co-registered to ST data using our Multi-Omics Imaging Integration Toolset (MIIT)[30]. After co-registration to ST data using MIIT spots that did not overlap with ST data were excluded. Supplementary Data 5 provides ion images of all metabolites pre- and post-co-registration used in this study.

### HRMAS NMR metabolomics

NOESY HRMAS NMR data was collected as previously described on a 600 MHz Bruker Avance III NMR spectrometer equipped with a magic angle spinning (MAS), $^1H/^{13}C$ probehead (Bruker Biospin)[21]. After automics phase[99] and baseline correction, spectra were probabilistic quotient normalized[100], aligned with icoshift[101] and ppm ranges specific for the respective metabolites (succinate: 2.415–2.395, lactate: 4.15–4.08, citrate: 2.74–2.47, glutamate: 2.36–2.305, taurine: 3.445–3.37) numerically integrated using the trapezoidal rule. For each metabolite the raw integrals were min-max scaled.

### Validation of prognostic power of APC and CEG signatures in large patient cohorts

To validate the APC and CEG signatures, we used open access cohorts with bulk gene expression data (cDNA microarray: 248 cases (56 with biochemical recurrence) from GSE116918[102] and RNA sequencing: 485 cases (99 with biochemical recurrence) from TCGA-PRAD[103] and a cohort, META855, of multiple case-cohorts (cDNA microarrays: GSE72291, GSE79957, GSE62116, GSE79915) of 855 cases (373 with biochemical recurrence, 85 with metastasis) including metadata as previously used to validate the Decipher prostate cancer genomic test[104]. Follow-up time and biochemical recurrence status for GSE116918 samples were directly extracted from respective entries in the metadata as published on NCBI Gene Expression Omnibus, last updated Oct 23, 2018. For TCGA-PRAD meta data were extracted from the follow-up and sample data (GDC Data Release v42.0). Only samples with database fields Tissue Type set to Tumor and Preservation Mode not set to FFPE were included. Samples with fields follow_ups.progression_or_recurrence set to Yes, follow_ups.progression_or_recurrence_type set to Biochemical, and follow_ups.days_to_follow_up and follow_ups.days_to_recurrence more than 8 weeks were included as cases with biochemical recurrence. Samples with follow_ups.progression_or_recurrence not set to Yes were included as cases

with no recurrence. Follow-up time in month for no recurrence cases was obtained by selecting the maximum value of follow_ups.days_to_follow_up and molecular_tests.days_to_test divided by 30.436875 days/month. Time to event in month for recurrence cases was derived by dividing the minimum value of follow_ups.days_to_follow_up and follow_ups.days_to_recurrence by 30.436875 days/month. For the resulting 3 cohorts with in total 1588 cases, we calculated the ssGSEA scores for the APC and CEG signatures, split the cases in each cohort into high, medium, and low tertiles, and generated the inverse Kaplan–Meier survival curves using the R-package survival (version 3.7-0) with biochemical recurrence (GSE116918, TCGA, and META855) and metastasis (META855). For GSE116918 and TCGA cohorts the hazard ratios were obtained by fitting a univariate Cox proportional hazards regression model using the coxph of the R-package survival (version 3.7-0). For the META855 cohort univariate and multivariate hazard ratios were obtained by fitting a Cox proportional hazards regression to a Fine-Gray competing risks model as previously described[104].

### Comparison of APC and CEG signature activity with prostate-related gene sets in public data

We used 41 prostate-related gene sets selected from various sources[9,10,24,52] in addition to our APC and CEG signatures for the ssGSEA analysis to evaluate prognostic power and relation to biological processes in the prostate of APC and CEG signatures (details in Supplementary Table S2). We used 38 of these gene sets for analysis in our ST and public data, and three additional gene sets for analysis only in public data (Supplementary Tables S2 and S3). The three additional gene sets were made by genes for three commercially available gene signatures for aggressive PCa, CCP, GPS and Decipher[31]. For these three, we used only the selection of genes from these signatures and not the scoring algorithms for scoring the signatures in clinical samples.

Prostate specific-cell type and immune-cell genes downloaded from Joseph et al. (Supplementary Tables S3 and S8)[10] contained genes enriched in five prostate-specific cell types, and prostate-specific enrichment of genes in 19 immune-cell types. To created prostate-specific cell type gene sets from these 24 cell types, we ranked the genes for each cell type by $q$-value (ascending) and fold-change (descending). The genes were sorted by the average rank over $q$-value and fold-change, and the top 75 ranked genes were selected to create a gene set for each cell type. We further validated each of the 19 immune-cell type gene sets with 142 corresponding immune gene sets downloaded from Azimuth (https://azimuth.hubmapconsortium.org/)[105], for PBMC (Peripheral Blood Mononuclear Cells) version 1-3, and Bone Marrow version 1 and 2. Each gene set from Azimuth contained 10 genes. We used GSEA[25] to score each of the 164 Azimuth gene sets in the 19 ranked immune gene sets from Joseph et al.[10], and then evaluated whether gene sets from Azimuth were enriched in our gene sets generated from Jospeh et al.[10]. The results generally showed enrichment of Azimuth immune gene sets with corresponding immune gene sets from Joseph et al., demonstrating that the gene sets downloaded from Joseph et al. represented their respective immune cell-types well. Two exceptions were CD4+ Naïve and CD8+ Naïve signatures which showed a more general enrichment in various CD4+ and CD8+ cells from Azimuth. The validation was also not able to distinguish subtypes of immune cells, for example different Dendritic subtypes (EREG, IGSF21, and TREM2), Basophil-Mast 1 and 2, Classical and Nonclassical Monocyte, and Natural Killer and Natural Killer T cells.

To compare these 41 gene sets and evaluate prognostic power and relation to biological processes in the prostate of our APC and CEG signatures in public data, we calculated ssGSEA scores for all gene sets in all 2512 samples from 12 public datasets (Supplementary Table S3). The public data gene expression values were not centered before ssGSEA analysis. Correlations between gene sets in each dataset were calculated by Pearson correlation. The overall correlation for two gene

sets was calculated as the average correlation over all 12 datasets. Complete tables with correlations for all gene sets in each individual dataset are given in Supplementary Data 2.

### Reporting summary

Further information on research design is available in the Nature Portfolio Reporting Summary linked to this article.

## Data availability

Bulk and spatial transcriptomics data are available upon request through Federated European Genome Phenome Archive (FEGA) Norway data access committee with accession number EGAC50000000277 and bundled under study EGAS50000000413. The metabolomics (HRMAS-NMR and MSI), stained images, and tissue annotations data are not externally archived as there are currently no suitable public data repositories that accept such sensitive data and that meet the data sharing criteria postulated by the study's ethical approval, patient consent, GDPR and Norwegian law. The metabolomics, stained images, and tissue annotations data can be requested via email to the corresponding authors. For all of the above datasets, access will only be granted after the following steps have been achieved; (1) the data requester and the intended use of the data must comply with GDPR regulation, Norwegian law, and the specific patient consent, (2) data sharing with the specific data requester must be approved by the regional ethical committee (REC) in Norway, (3) the Data Protection Impact Assessment (DPIA) may require revision and (4) there must be a signed data transfer agreement between the institution of the data requester and NTNU. Depending on the intended use of the data, the data requester can also be required to establish a collaboration agreement with NTNU prior to data sharing. Public data used available under: GSE116918, GSE72291, GSE79957, GSE62116, GSE79915, GSE8218, GSE21034, GSE16560, GSE46691, GSE70768, GSE70769, GSE46602, GSE32571, GSE97284, GSE172357, GSE141445, TCGA-PRAD (https://portal.gdc.cancer.gov/projects/TCGA-PRAD), GTEx-Prostate (https://www.gtexportal.org/home/downloads/adult-gtex/bulk_tissue_expression#bulk_tissue_expression-gtex_analysis_v8-rna-seq), E-MTAB-1041 (https://www.ebi.ac.uk/biostudies/ArrayExpress/studies/E-MTAB-1041). Source data are provided with this paper.

## Code availability

Code for pre-processing of the spatial transcriptomics data is available under https://doi.org/10.5281/zenodo.13912230 and https://doi.org/10.5281/zenodo.16738839.

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

## Acknowledgements

This research was funded by the European Research Council (ERC) under the European Union's Horizon 2020 research and innovation program (grant agreement no. 758306, awarded to M.B.T.), Norwegian University of Science and Technology (NTNU), the Liaison Committee between the Central Norway Regional Health Authority (RHA) and NTNU, the Norwegian Cancer Society (grant 272770 awarded to M.K.A., grant 208263-2019 awarded to M.B.T.) and Terje Eugen Johnsen funds (awarded to M.B.T.). May-Britt Tessem's research project is part of the Centre for Digital Life Norway, which is supported by the Research Council of Norway's grant 248810. All tissue samples were collected and stored by Biobank1, St. Olav's Hospital. Tissue sectioning, staining and scanning were performed by or in collaboration with the Histology lab at the Cellular & Molecular Imaging Core Facility (CMIC) at NTNU. The authors thank Ingunn Nervik, Kathrin Juanita Gravvold Torseth, and Borgny Ytterhus for excellent technical assistance during these experiments. Biobank1 performed isolation of RNA before bulk analysis. Transcriptomics experiments were carried out at the Genomics Core Facility at NTNU. MALDI MSI and HRMAS NMR acquisitions were achieved using instrumentation at the MR Core facility, NTNU. A.U. wishes to thank the Research Council of Finland project no. 349314; Cancer Foundation Finland; Norwegian Cancer Society project no. 198016-2018 & project no. 273672 –2023, Tampere Institute for Advanced Study.

## Author contributions

Conceptualization: S.K., M.K.A., M.B.T., M.B.R., and A.U.; Data curation: S.K., M.K.A., E.M.S., M.W., and M.B.T.; Formal analysis: S.K., M.K.A., M.W., A.K., A.S., S.H., Y.H., M.H., T.V., Ø.S., and M.B.R.; Funding acquisition: M.B.T.; Investigation: S.K., M.K.A., E.M.S., M.W., T.V., Ø.S., and M.B.T.; Methodology: E.M.S., S.K., M.K.A., M.W., and M.B.T.; Project administration: M.B.T., A.U., and M.B.R.; Resources: E.D., R.J.K., D.E.S., and M.B.T.; Supervision: E.D., G.F.G., M.N., M.B.R., A.U., and M.B.T.; Writing—original draft: S.K., M.K.A., and M.B.T.; Writing—review & editing: all authors contributed.

## Funding

 Olavs Hospital - Trondheim University Hospital).

## Competing interests

M.A. and E.D. are employees and Y.H. is a former employee of Veracyte, Inc (San Diego, CA) who make the Decipher prostate genomic classifier test and own the GRID software/database. The authors declare no further competing interests.
