## [Transparent Peer Review file · Nature Communications]

Spatial multi-omics identifies aggressive prostate cancer signatures highlighting pro-inflammatory chemokine activity in the tumor microenvironment

Corresponding Author: Dr Sebastian Krossa

Version 0:

Reviewer comments:

Reviewer #1

(Remarks to the Author)

Summary

The research paper explores the molecular characteristics of the tumor microenvironment in aggressive prostate cancer, identifying relapse-associated (RA) and chemokine-enriched gland (CEG) signatures. These signatures are linked to immune response, chemokine expression, metabolic changes, and potential biomarkers in prostate cancer progression. The study emphasizes the importance of spatial transcriptomics, histopathology integration, and multi-omics approaches in understanding tumor heterogeneity and identifying therapeutic targets. This study also shows following novel points: 1) Integration of multi-layered spatial analysis, immunohistochemistry, and mass spectrometry imaging to provide a comprehensive understanding of the tumor microenvironment and metabolic changes; 2) Association of RA and CEG signatures with loss of luminal features and citrate secretion in glands, suggesting a transition from a healthy prostate phenotype to a molecular phenotype of prostate cancer; and 3) Correlation of RA and CEG signatures with specific cell populations, particularly immune cells and Club-like cells, indicating their potential role in tumor progression and relapse. There are some issues that should be addressed by the authors.

Major critiques

1. The RA and CEG signature were not validated with a large set of prostate cancer cohort with transcriptome data.
2. Many previous studies provided relapse associated genes and immune phenotypes. However, this study does not provide any reliable comparison with the previous signatures and immune phenotypes.
3. Although this study performed comprehensive multi-omics molecular profiling, it does not have enough level of depth in patients and tumor phenotyping such as lifestyle, environmental factors, genetic background, and other clinical factors.
4. It would be interesting to check the expression of these genes in enzalutamide-resistant cell lines and see if knockdown of key genes such as ACKR1 results in functional changes.
5. Further systematic comparison with luminal cells seems to be necessary to confirm the correct identification of club-like cells.

Minor critiques

1. The first paragraph describing the study cohort in Result section should go to Method section.

Reviewer #2

(Remarks to the Author)

Review of Nature Communications

This manuscript presents some fascinating data obtained using a range of platform technologies applied initially to 8 cancerous prostates: the technologies include spatial transcriptomics (Visium 10X), metabolite detection using MALDI (TOF)MSI, and HRMAS NMR metabolomics data. These are combined with a number of standard approaches including histopathology and IHC as well as bioinformatic investigations. Integrating these approaches has generated a rich dataset corresponding to 19854 circular spots from 32 tissue samples from 8 prostate cancer patients. The cancers were selected

based on whether they had undergone relapse (5 patients) or not (3 patients).

1. There is an issue in relation to the description of the prostatectomy samples. The schematic shown in Figure 1a shows a prostate slice containing a single prostate cancer region. This is unusual because most prostate cancer is multifocal with two or more independent areas of cancer in a single slice. Actual maps of all eight selected prostates needs to be shown as a Figure in the Supplementary material; in each case showing the regions of cancer and other areas selected. If the authors have deliberately selected prostates with a single area of cancer in each case they need to explain why they have made this choice. Of course when there are two or more independent areas of cancer (as is common) then it is not possible to tell which of these is relevant to cancer progression; there are papers documenting that in some cases it is the smaller region of cancer that progresses. This all needs discussing in the text. What this means is that the initial dataset on 8 patients can, at best, be considered as a hypothesis generating data set.

2. There is an issue in relation to the clinical description of the two groups of cancerous prostates: relapsed and non-relapsed. The definition in the manuscript of relapsed patients is as follows: biochemical recurrence (PSA > 0.1 ng/ml) and/or confirmed metastasis. However, there is a big difference between PSA failure and metastasis particularly at 10 years. The authors need to mention in the main text how many of the five selected cancers are in each group. They also need to give a justification for this selection.

3. There is a critical problem in the overall design of the study. The manuscript was well written and easy to follow. However, there was little statistical power both in the hypothesis generating set (8 patients) and in the validation set (37 patients). There was also an overlap between these two set (the initial 8 patients are include entirely within the 37 validation cohort of patients). This design completely invalidates any statistical meaning irrespective how many samples are taken from each patient.

4. I found the arguments in the body of the text quite convincing based on the data presented. However, at best, because of the small number of patients, they can only be considered anecdotal, generating a hypothesis that can be tested in a much larger dataset. Unfortunately, the slightly larger dataset presented overlapped with the discovery dataset and in any case was not big enough. To provide a really convincing confirmation a larger dataset of hundreds of patients would be required. In this reviewers experience there are many cases where biomarkers found to be statistically significant in smaller dataset have not been confirmed in much larger dataset (100s of patients). I would encourage the authors to take the time to carry out this confirmation, as the results of their study, if true, are significant. I wonder if a number of publicly available datasets could be used; confirmation in multiple publicly available datasets would make the results very strong. Confirmation on a much larger dataset would be needed if the work is to be published here. In any case the confirmation dataset would need to be entirely independent of the smaller hypothesis generating dataset.

5. The manuscript presents a number of sets of analysis after selecting of the original cancers, not cancer areas and the resulting spots. They first generate a signature called RA that contains many immunologically related genes. A CEG signature was then calculated that at a single spot level (ssCEG) showed enrichment in all classes of spots (cancer, field effect, normal). This was OK but is was not clear why the spots located immediately adjacent to the cancer were designated "field effect". This implies knowledge of mechanism. The authors should choose something more neutral like "morphologically normal tissue adjacent to cancer".

6. In the next section the authors looked at the RA and CEG in relation to luminal status and citrate secretion. Here I was unconvinced by the rational of fusing cancer and field effect sample; if there were a low number of field effect samples as stated then these should be excluded to make a cleaner comparison between cancer and normal. Indeed you could argue that the field effect samples should be merged with the normal samples, since they are both non-cancerous. This section needs reworking.

7. The authors next present evidence that RA and GEG signatures are associated with Club-like cells and immune cell enrichment. This section provides some very interesting anecdotal observations that need confirming in much larger independent datasets. In the following section the authors headline that the RA and CEG signatures arise from Club-like cells. The arguments were a bit tenuous and not as convincing as other aspects of the paper. Based on the correlation evidence presented the authors could conclude that an origin in club-like cells is one of the options so the work "likely" in the title is overstating the case. This section would need modifying so that conclusions from the data are not overstated. Other sections would need modifying also. Correlations need to be presented in a more "lay friendly" way. " $p \geq 0.25$ " does not mean much to most people; what is the statistical significance? Please rework this.

8. The next section presents a case that Atypical Chemokine Receptor 1 (ACKR1) is the dominant receptor for the CEG signature. This conclusion seemed to be based on the observation that only expression of CXCR4 and ACKR1 was detected in the ST data. This section was more convincing but again there needs to be a more lay friendly presentation of statistically significant correlations. The remaining sections were also convincing.

9. Discussions were a bit long winded and I recommend shortening by 50% or so for a final manuscript. There seemed to be a lot of unnecessary detail. Just keep to key discussion points.

Overall this was an interesting paper that contained potentially significant novel observations. However many observations were anecdotal only. The study design had critical flaws that need addressing before the paper could be accepted.

Reviewer #3

(Remarks to the Author)

In the manuscript titled "Deep Phenotyping of the Prostate Tumor Microenvironment Reveals Molecular Stratifiers of Relapse Linked to Inflammatory Chemokine Expression and Aberrant Metabolism" Sebastian Krossa et al. apply spatially resolved profiling methods to determine molecular features of prostate cancer foci with inferred aggressive behavior. Broadly, risk stratification for localized prostate cancer represents a major unmet medical need. While numerous expression signatures have been introduced in recent years aimed at addressing these needs, this study takes a different approach by applying spatial transcriptomics and metabolomics profiling to primary tumor specimens. In a relatively limited set of samples, the authors find that gene expression signatures associated with inflammation and a previously defined epithelial cell type ("club-like cells") show an association with disease behavior.

While this manuscript represents an interesting descriptive study of a small number of specimens, it lacks focus and clear clinical relevance. There are several other points to consider:

1. The number of samples in the discovery set is very low, and extensive external validation in datasets with long-term follow-up is missing.
2. The identification of "benign" areas with cancer-associated changes is challenging to contextualize given the relatively low resolution of the method used. In situ validation and documentation of this finding through detailed histologic analysis would be necessary.
3. The general theme that inflammation in the prostate is associated with carcinogenesis and adverse clinical features is, in itself, not novel. There is substantial literature on putative inflammatory precursors, such as PIA, that show cancer-associated changes.

Reviewer #4

(Remarks to the Author)

This study identified two immune-associated signatures, characterized by increased chemokine production and dysregulated citrate metabolism, that are selectively linked to aggressive prostate cancer via integration of spatial multi-omics (transcriptomics and metabolomics). Notably, the authors reported that this occurred even in non-cancerous glands of patients who relapsed after surgery, and further speculated that immune infiltration and inflammation in the tumor microenvironment may be a driver of progressive disease. The manuscript also suggests the potential utility of chemokines as biomarkers of relapse, with linkages to metabolism through modulation of citrate.

Positive aspects of the study include the use of multiple sampling sites in each patient tumor to incorporate intratumoral heterogeneity and the microenvironment, and that the authors excluded prostatitis as a potential, though rare, confounding factor in the gene signatures. However, the study design fundamentally compromises interpretation of the study's findings, particularly concerning patient selection in a very small cohort. The reported outcomes therefore currently do not adequately support the authors' conclusions. Additionally, the authors need to address several major concerns, which are detailed in the review below.

Comments:

1. In Supplementary Table 1, column 3, 'Time to relapse (months)', could the authors clarify the '0' values for a subset of relapse patients? Currently, it appears that '0' means patients who 'relapsed' actually were cases of non-curative surgery where PSA levels did not decrease to undetectable levels and, therefore, depending on the definition, did not represent a true relapse – these patients are already known to have poorer outcomes, likely due to pre-existing metastatic spread or locoregional disease. Notably, 4 out of 5 patient samples used for spatial transcriptomics had the '0' time to relapse, which is not an ideal set of samples for this comparative study, and 11 out of the 27 broader cohort. The tumor grades across patients were imbalanced and particularly higher for patients who 'relapsed', and 4 of the relapsed patients had pre-operative PSA of over 30. Given the constraints of cost in undertaking this type of multi-omic approach, which necessitates smaller sample numbers, it is even more important to control for these clinical factors as much as possible and include only patients who experienced an apparently curative surgery. I believe the authors characterize a completely different biology here, based on the stage of disease progression rather than a signature of relapse.
2. In the discussion, the authors briefly described the function of each gene derived from the immune-associated signatures and their known role(s) in cancer, but do not elaborate on the biology that is uncovered. Is there a distinct population of immune cells driving this phenotype? What are the drivers of this response in non-cancerous cells? Could the authors also perform cell-type deconvolution on the bulk transcriptomics data to confirm this observation in the spatial data?
3. Concerning the discrepancy observed in the comparison between spatial transcriptomics and bulk transcriptomics data, how representative are the patient tissues used for bulk transcriptomics compared to the patient tissues used for spatial omics? Based on the table, it looks like tissues used for bulk transcriptomics had higher grades of tumors, but this is not accounted for.
4. Could the authors expand and validate the observed immune signatures in a spatial context in an independent cohort of patient tissues, possibly addressing the issues of clinical relapse discussed in point 1 above?
5. In Figure 2A, the localization of the signature signals does not appear to very clearly match their pathological features (e.g., NCG). It would be helpful to include H&E images for each tissue in this figure to show how well the ST deconvolution reflects the histopathology and vice versa.
6. Were the authors able to determine the association of signatures with progression-free survival in publicly available bulk datasets? Looking at the datasets used in Supplementary Table 3, some of the datasets (ie. Taylor and Sboner) have outcome data. The authors could try to compare their signatures in more aggressive prostate cancer samples (ie. metastatic samples) rather than just normal vs cancer.

7. For the metabolomics data, the data are not clearly described in the text and the biological significance is not clear – was the metabolism perturbed in a specific cell-type? Besides the correlation between gene and metabolites of citrate metabolism, were there other noteworthy correlations at both the transcriptional and metabolic level?

8. There are known ambiguities in identifying metabolites using untargeted mass spectrometry imaging—there is no mention of the m/z or ppm/mDa range in the manuscript. In their methods section, “metabolites were identified by MS/MS and accurate masses from our previous study using the same methods for equivalent PCa samples”. Was MS/MS performed on these specific patient samples or inferred from the previous study? When making significant claims about correlations between analytes, and especially in this study their relationship to disease progression, MS/MS should ideally be on these exact samples or if not possible, this should be discussed as a study limitation.

9. Notably, the authors did not present any spatial metabolomics data in their main figures or supplementary figures, which is an important omission. In Figure 5, including only box plots without any images or spectra is not showing the data, and while alterations in metabolite abundance are claimed based on these box plots, this is unconvincing and no statistical proof is provided. Representative m/z images of each metabolite should at a minimum be shown – boxplots are informative but can be misleading. For example, comparing two ROIs between tissue sections can show a significant increase in a mass abundance that, when compared to the image, can reveal a background or matrix ion that, by chance, has higher intensities in one of the ROIs compared to the other.

Minor comments:

1. In Supplementary Figure 4, high RA signature activity was mostly detected in the stromal regions of the tissue (based on histopathology). However, the authors claim enrichment of club-like epithelial cells associated with the RA signature.

2. In Supplementary Table 1, the patients with ‘control’ status had months indicated for ‘time to relapse’. As these patients did not by definition relapse, for clarity this column should indicate N/A with the number of months actually reflecting the follow up time rather than time to relapse.

3. For the signature generation, was there a reason for selecting only the top 5 differentially expressed genes? Did the authors try to evaluate the robustness of the signatures?

Reviewer #5

(Remarks to the Author)

Version 1:

Reviewer comments:

Reviewer #1

(Remarks to the Author)

The authors have adequately addressed the issues of all of the reviewers to my satisfaction and the submission is ready for publication from my perspective.

Reviewer #2

(Remarks to the Author)

This manuscript is significantly improved relative to the original. Mainly because of the confirmation series. I would be happy for it to be accepted for publication

Reviewer #3

(Remarks to the Author)

The authors have addressed most of my comments. The addition of substantial new clinical data has greatly strengthened the paper.

Reviewer #4

(Remarks to the Author)

The authors have substantially revised their manuscript by incorporating additional analyses, clarifying key aspects of the methodology, and adding new supporting data. Most importantly, they have more accurately classified the patient groups with respect to relapse, validated their findings in multiple independent cohorts, and provided the requested spatial metabolomics data with relevant images and spectra. Fundamentally, much of the data remains descriptive and hypothesis-generating rather than conclusive. However, many of my concerns related to data presentation and analytical approach have been addressed.

Reviewer #5

(Remarks to the Author)

Response to reviewers' comments on the article "Deep phenotyping of the prostate tumor microenvironment reveals molecular stratifiers of relapse linked to inflammatory chemokine expression and aberrant metabolism" submitted to Nature Communications under the manuscript ID NCOMMS-24-43634-T

We thank the reviewers for their careful reading and for providing important and relevant comments to our manuscript. The major remark was a lack of validation of the Aggressive Prostate Cancer -APC (former RA) and CEG signatures which we have now performed on 8 different patient cohorts representing altogether 1588 patients. Due to updated clinical data on the ST data, we renamed the relapse signature (RA) to APC based on the heterogeneous endpoints in the aggressive disease group, including biochemical recurrence, onset of persistent and metastatic disease. APC is predictive and prognostic for all endpoints.

The results are presented in the improved version of our manuscript, and we have responded in blue to all the comments of the reviewers.

Reviewers' comments:

Reviewer #1 (Remarks to the Author):

Summary

The research paper explores the molecular characteristics of the tumor microenvironment in aggressive prostate cancer, identifying relapse-associated (RA) and chemokine-enriched gland (CEG) signatures. These signatures are linked to immune response, chemokine expression, metabolic changes, and potential biomarkers in prostate cancer progression. The study emphasizes the importance of spatial transcriptomics, histopathology integration, and multi-omics approaches in understanding tumor heterogeneity and identifying therapeutic targets. This study also shows following novel points: 1) Integration of multi-layered spatial analysis, immunohistochemistry, and mass spectrometry imaging to provide a comprehensive understanding of the tumor microenvironment and metabolic changes; 2) Association of RA and CEG signatures with loss of luminal features and citrate secretion in glands, suggesting a transition from a healthy prostate phenotype to a molecular phenotype of prostate cancer; and 3) Correlation of RA and CEG signatures with specific cell populations, particularly immune cells and Club-like cells, indicating their potential role in tumor progression and relapse. There are some issues that should be addressed by the authors.

Major critiques

1. The RA and CEG signature were not validated with a large set of prostate cancer cohort with transcriptome data.

We thank the reviewer for this comment which allowed us to improve substantially the confidence in our results. We have now performed additional analyses on both the RA, renamed to aggressive prostate cancer (APC), and the CEG signatures in open access cohorts (microarrays: GSE116918 and RNA seq: TCGA-PRAD) and a larger patient cohort (META855) in collaboration with the company Veracyte

(Spratt et al, JCO 2017). We validated APC based on various endpoints in the aggressive group, including biochemical recurrence and onset of persistent and metastatic disease. APC is predictive and prognostic for all endpoints.

META855 is a cohort of multiple case-cohorts of 855 patients that was previously used to validate the Decipher prostate cancer genomic test. Altogether, all the validation cohorts included approximately 1588 prostate cancer patients. The validation cohorts are described in the Methods on page 27:

“Validation of prognostic power of APC and CEG signatures in large patient cohorts

To validate the APC and CEG signatures, we used open access cohorts with bulk gene expression data (cDNA microarray: 248 cases (56 with biochemical recurrence) from GSE116918 [101] and RNA sequencing: 485 cases (99 with biochemical recurrence) from TCGA-PRAD [102] and a cohort, META855, of multiple case-cohorts (cDNA microarrays: GSE72291, GSE79957, GSE62116, GSE79915) of 855 cases (373 with biochemical recurrence, 85 with metastasis) including metadata as previously used to validate the Decipher prostate cancer genomic test [104]. Follow-up time and biochemical recurrence status for GSE116918 samples were directly extracted from respective entries in the metadata as published on NCBI Gene Expression Omnibus, last updated Oct 23, 2018. For TCGA-PRAD meta data were extracted from the follow-up and sample data (GDC Data Release v42.0). Only samples with database fields ‘Tissue Type’ set to ‘Tumor’ and ‘Preservation Mode’ not set to ‘FFPE’ were included. Samples with fields ‘follow_ups.progression_or_recurrence’ set to ‘Yes’, ‘follow_ups.progression_or_recurrence_type’ set to ‘Biochemical’, and ‘follow_ups.days_to_follow_up’ and ‘follow_ups.days_to_recurrence’ more than 8 weeks were included as cases with biochemical recurrence. Samples with ‘follow_ups.progression_or_recurrence’ not set to ‘Yes’ were included as cases with no recurrence. Follow-up time in month for no recurrence cases was obtained by selecting the maximum value of ‘follow_ups.days_to_follow_up’ and ‘molecular_tests.days_to_test’ divided by 30.436875 days/month. Time to event in month for recurrence cases was derived by dividing the minimum value of ‘follow_ups.days_to_follow_up’ and ‘follow_ups.days_to_recurrence’ by 30.436875 days/month. For the resulting 3 cohorts with in total 1588 cases, we calculated the ssGSEA scores for the APC and CEG signatures, split the cases in each cohort into high, medium, and low tertiles, and generated the inverse Kaplan-Meier survival curves using the R-package survival (version 3.7-0) with biochemical recurrence (GSE116918, TCGA, and META855) and metastasis (META855). For GSE116918 and TCGA cohorts the hazard ratios were obtained by fitting a univariate Cox proportional hazards regression model using the coxph of the R-package survival (version 3.7-0). For the META855 cohort univariate and multivariate hazard ratios were obtained by fitting a Cox proportional hazards regression to a Fine-Gray competing risks model as previously described [104].”

The high APC scoring tertile had a significantly increased risk for biochemical recurrence in all validation cohorts. In addition, APC high scoring patients had also an increased risk of metastasis. High CEG signature score associated with increased risk of biochemical recurrence in validation cohorts, however only statistically significant in the GSE116918 cohort. This convincingly validates the predictive power of the APC signature highlighting the usefulness of utilizing spatial information to derive robust bulk gene expression signatures and confirms that the CEG signature is rather a spatial signature within the context of the prostate cancer microenvironment indicating potentially aberrant prostate glands. Due to the potential large degree of convolution within bulk transcriptome data, we were not able to validate the predictive and prognostic power of CEG signal which perhaps requires larger cohorts of spatial transcriptome data of primary prostate cancer with follow up data.

The data are presented in a new Figure 6 and new text about the validation results are inserted in the manuscript on page 17

“APC signature is predictive for aggressive PCa in large public bulk data sets

To test the clinically predictive power of APC and CEG signatures, we performed survival analysis in 3 publicly available cohorts and a cohort of multiple case-cohorts (META855, previously used to validate the Decipher prostate cancer genomic test) comprising a total of 1588 prostate cancer patients. The inverse Kaplan-Meier plots (Figure 6) demonstrate that the APC signature is predictive for biochemical recurrence in all cohorts with significantly increased hazard ratios (HR) for patients with high APC scores. Further, using the META855 cohort, the APC signature was also predictive for metastasis with a significantly increased HR (Figure 6g). Of note, using the meta data associated with the META855 cohort we obtained significantly increased HR using a multivariate Fine-Gray subdistribution hazard model for high APC scoring patients for biochemical recurrence and not for the multivariate but the univariate model for metastasis (Supplementary Tables 4-9). The HR for biochemical recurrence for patients scoring high in CEG was increased in all validation cohorts, however only significant for the GSE116918 cohort.”

Further, we edited the abstract, discussion and conclusion to reflect these new findings.

2. Many previous studies provided relapse associated genes and immune phenotypes. However, this study does not provide any reliable comparison with the previous signatures and immune phenotypes.

Many different genes have been associated with prostate cancer relapse in numerous studies, but it is no clear consensus on what is regarded as well-established markers for relapse. Three gene signatures (CCP, GPS and Decipher (PMID: 26975490)) make up the current available clinical tests for aggressive prostate cancer. Of the total 61 genes used in these tests, only one gene is shared between any of the three tests (*NUSAP1* in CCP and Decipher), illustrating the lack of consensus. Both the APC and CEG signatures did not overlap with any of the 61 genes but were still validated in external datasets. To focus specifically on immune genes, we also checked a signature of 12 immune related genes that were shown to associate with relapse in a study by Lv et al., (reference in Table 1 in this document) but neither of these genes overlapped with the APC or CEG signatures. Only one of the 12 immune genes (*TPX2*), overlapped with the three clinically available signatures. However, *TPX2* is better known as a cell-cycle regulator during mitosis rather than an immune gene. Further, we investigated 2 recently published metabolism-related and 4 PCa predictive signatures (details in Table 1) but did not find any overlap with our APC or CEG signature genes. In summary, previous studies show substantial variation in identified genes, and it is not surprising that signatures derived from different types of data such as spatial transcriptomics will generate new signatures related to prostate cancer aggressiveness. Such signatures would be more targeted to tissue compartments compared to signatures generated on bulk data.

We understand our approach to deconvolute the ST data using sc-RNAseq data as inclusive of immune phenotyping. Our Figure 3 shows derived cell type compositions including detailed immune phenotypes. Figure 3b shows correlation with our signatures and these results are discussed in the respective sections on page 9-10 in the main text. Further, using public data we compare our signature to multiple immune cell signatures and the PCa signatures CCP, GPS, and Decipher in Figure 7.

We added to the discussion on page 18 to clearly state that there is no overlap of the genes with other signatures for PCa:

“The genes in APC and CEG do not overlap with clinically or commercially used signatures (such as CCP-Prolaris, GPS-Oncotype Dx, Decipher, or My Prostate 2.0-Lynx Dx) or recently published PCa biomarker panels [31-35].”

Table 1 Overview PCa gene panels compared with APC and CEG signature genes

Signature Panel Name	DOI Publication or Source
CCP / Prolaris	10.4103/1008-682X.175096
GPS / Oncotype Dx	10.4103/1008-682X.175096
Decipher	10.4103/1008-682X.175096
Lv et al. 2021	10.1007/s00262-021-02923-6
Li et al. 2022	10.3389/fimmu.2022.982628
Hu et al. 2024	10.1186/s40001-024-01672-3
Shen et al. 2024	10.1038/s41416-024-02854-w
Mercola 7-gene signature for prostate cancer	NIH Early Detect Research Network, phase 3
My Prostate 2.0 genes	Commercial test Lynx Dx
Chinnaiyan (Laxman) 7 marker panel for prostate cancer	NIH Early Detect Research Network, phase 1

3. Although this study performed comprehensive multi-omics molecular profiling, it does not have enough level of depth in patients and tumor phenotyping such as lifestyle, environmental factors, genetic background, and other clinical factors.

The tissue from patients included in this study were retrieved from our local hospital biobank (Biobank1, St Olav University Hospital) established back in 2008. Patients were chosen based on various clinical factors; large enough cancer (to extract two 3 mm cores) found within the 2 mm thick whole prostate tissue slice, recurrence status, no additional adjuvant treatment and a patient with longest follow-up time possible. All other clinical parameters are reported in Supplementary Table 1. We had no access to any lifestyle data except the patient’s hospital record associated with the clinical follow-up data of potential treatment after surgery and mortality. The patients used in this cohort had no records of additional genetic background.

The added validation of our signatures using in total 1588 cases, significantly increases the level of depth in patients. Increasing the level of depth in lifestyle, environmental factors, genetic background, and other clinical factors would be interesting but is unfortunately not feasible as such data is often not readily available or collected in a harmonized fashion to allow meaningful analysis. Nevertheless, the added cohort META855 adds a broader range of ethnicity and clinical parameters as reported in Supplementary Tables 4-9.

4. It would be interesting to check the expression of these genes in enzalutamide-resistant cell lines and see if knockdown of key genes such as ACKR1 results in functional changes.

We thank the reviewer for this comment which allowed us to perform additional analysis. We have not performed ACKR1 knockdown in cell lines as the expression of this gene in the cancer model seems to be very low. We have initially analysed cell lines data and clinical data and ACKR1 is listed as DARC in the Alumkal et al., dataset (doi: 10.1073/pnas.1922207117) of ENZA responsive and non-responsive prostate cancer patients with very low expression and no significant difference between high and low expression groups (the group assigned either based on the median expression or the top and bottom

quartiles, see Figure 1 a and b for survival plots). Unfortunately, we did not find ACKR1 in the LNCaP parental and enzalutamide-resistant cell lines single-cell data (Taavitsainen et al., 2021, doi: 10.1038/s41467-021-25624-1). Puzzled by these findings, we also analysed the expression of ACKR1 in bulk gene expression data from the same LNCaP parental and enzalutamide-resistant cell lines (Handle et al., 2019, doi: 10.1038/s41598-019-50220-1), and the expression of ACKR1 was undetectable in this dataset too. We could however detect the expression of this gene in the gene expression data from the SU2C prostate cancer cohort (doi: 10.1073/pnas.1902651116). Enzalutamide and Abiraterone naïve patients were divided according to ACKR1 high and low expressing tumors, and a Kaplan-Meier was plotted showing no significant difference in survival of ACKR1 low and high patients (Figure 1 c and d). The separation of the curve trend improved comparing the highest and lowest quartile although the differences between the curves were not statistically significant.

Figure 1 Survival analysis of ACKR1 high vs low expression groups. (a & b) Kaplan-Meier plots based on ACKR1/DARC expression in Alunkal et al. dataset. (c & d) Kaplan-Meier plots based on ACKR1/DARC expression in SU2C prostate cancer cohort. Grouping into high (light red) and low (cyan) ACKR1 expression groups using (a & c) top and bottom quartiles or (b & d) media quartiles.

ACKR1 (previous official gene name: DARC) regulates neutrophil counts in blood and, in particular, it is expressed in nucleated erythrocytic cells in the bone marrow. This protein likely regulates the flux of these cells through chemokine interaction (doi: 10.3389/fimmu.2023.1111960). We have therefore tested expression of ACKR1 in Dong et al (CRPC, Figure 2 left, doi: 10.1038/s42003-020-01476-1) and Chen et al (high-grade PC, Figure 2 right, doi: 10.1038/s41556-020-00613-6) prostate cancer single cell datasets, and, indeed, ACKR1 has very low expression in all cell types annotated but is present in the endothelial cells. This result has been included in the article on page 11 and Figure 2 was integrated into Supplementary Figure 8:

“To confirm the cell type-specific expression pattern of ACKR1, we analyzed scRNA-seq data from prostate tumors and castrate-resistant prostate cancer (CRPC) needle biopsies [27, 28]. Our analysis

confirmed that *ACKR1* is predominantly expressed by endothelial cells (Supplementary Figure 8b). Additionally, at the CRPC stage, a subset of club-like and basal cells also exhibited *ACKR1* expression.”

Figure 2 Expression of *ACKR1* receptor in different cell types derived from scRNAseq data from prostate tumors (Cheng, doi: 10.1038/s41556-020-00613-6) and CRPC needle biopsies (Dong, doi: 10.1038/s42003-020-01476-1) visualized as dot-plots representing the fraction of cells in each group by diameter and the normalized mean expression of *ACKR1* by color.

We conclude that the *ACKR1* is not expressed in cancer cells but in endothelial cells (see also Figure 4c in the manuscript showing similar results from our spatial transcriptomics analysis), and this justifies the undetectable expression of *ACKR1* in the cell lines data.

The expression levels of *ACKR1* in gene expression data from clinical samples are varying i.e. in Taylor et al, 2010 (doi: 10.1016/j.ccr.2010.05.026), the expression is lower in metastatic samples and in cell lines (LNCaP line is included in this dataset) compared to primary cancers (Figure 3).

Taylor et al. (2010) - Integrative Genomic Profiling of Human Prostate Cancer

Gene expression boxplot

Figure 3 DARC (*ACKR1*) expression in cell lines vs clinical samples as published by Taylor et al., 2010 (doi: 10.1016/j.ccr.2010.05.026).

In Figure 4 we report correlation of expression of *ACKR1* with endothelial cells. We reworded the sentence stating this result in the main text (page 11):

“The strongest and single distinct positive correlation of ACKR1 was observed with endothelial cells ($\rho = 0.29$) and the strongest negative correlation with luminal epithelial cells ($\rho = -0.12$).”

Therefore, we believe that the CEG-signature reflect some features of these tumors to extravasate or attract relevant immune infiltrating cell types.

5. Further systematic comparison with luminal cells seems to be necessary to confirm the correct identification of club-like cells.

We thank the reviewer for this important comment. We have performed this comparison thoroughly and more focused in our recent publication: Kiviaho A et al. Single cell and spatial transcriptomics highlight the interaction of club-like cells with immunosuppressive myeloid cells in prostate cancer. Nat Commun. 2024 Nov 16;15(1):9949. doi: [10.1038/s41467-024-54364-1](https://doi.org/10.1038/s41467-024-54364-1). We have referenced the article now also in this manuscript in the introduction on page 2:

“Through spatial transcriptomics analysis we recently demonstrated that Club-like cells are a key epithelial cell subtype associated with myeloid inflammation [20].”

And in the discussion on page 19: *“This shows further evidence on the important role of Club-like cells in immunological processes in the prostate that we recently uncovered [20].”*

Minor critiques

1. The first paragraph describing the study cohort in Result section should go to Method section.

Considering the nature-typical layout of “Results section first” we opted to include this information to the results part to help the reader to better understand the study design. We are open to move this part and leave the decision with the Editors of the journal.

Reviewer #2 (Remarks to the Author):

Review of Nature Communications

This manuscript presents some fascinating data obtained using a range of platform technologies applied initially to 8 cancerous prostates: the technologies include spatial transcriptomics (Visium 10X), metabolite detection using MALDI (TOF)MSI, and HRMAS NMR metabolomics data. These are combined with a number of standard approaches including histopathology and IHC as well as bioinformatic investigations. Integrating these approaches has generated a rich dataset corresponding to 19854 circular spots from 32 tissue samples from 8 prostate cancer patients. The cancers were selected based on whether they had undergone relapse (5 patients) or not (3 patients).

1. There is an issue in relation to the description of the prostatectomy samples. The schematic shown in Figure 1a shows a prostate slice containing a single prostate cancer region. This is unusual because most prostate cancer is multifocal with two or more independent areas of cancer in a single slice. Actual maps of all eight selected prostates needs to be shown as a Figure in the Supplementary material; in each case showing the regions of cancer and other areas selected. If the authors have deliberately selected prostates with a single area of cancer in each case they need to explain why they have made this choice. Of course when there are two or more independent areas of cancer (as is common) then it is not possible

to tell which of these is relevant to cancer progression; there are papers documenting that in some cases it is the smaller region of cancer that progresses. This all needs discussing in the text. What this means is that the initial dataset on 8 patients can, at best, be considered as a hypothesis generating data set.

We thank the reviewer for the important comment and the suggestion. Figure 1a simply serves illustrative purposes showing an example of how we extract cores according to pathology where 2 cores were taken from slice/prostate that usually had one tumor focus, one core out of the tumor and one core from the supposedly normal tissue portion. We have added the file “Supplementary_File_4_ST_cores_sampling_sites.pdf” to the supplementary file, showing each individual sampling site of the samples from the 8 patients used in the spatial analysis.

We appreciate the point brought up by the reviewer about the multifocality nature of some prostate cancer patients. We analysed multi-foci gene expression data from Strømme et al. 2022 (doi: 10.1038/s41417-022-00444-7) to understand the intra-patient variability of the APC and CEG signatures and potential implications of multifocality (Figure 4). The dataset was missing data on 6 of the 28 APC signature genes limiting interpretability. However, we observed that for most cases the intra patient variability was limited for both signatures and lower than the overall variability reflecting the finds from our own multi-sample bulk data (Supplementary Figures 9 & 10). Due to the missing genes we decided to not include Figure 4 into the manuscript. However, we touch upon the topic of multifocality in the discussion on page 21:

“Further, despite our study design not consequently focusing on multi-foci sampling, the APC signature displayed relatively low intra-patient variability for most of the patients in our own multi-sample and partly multi-foci bulk data (Supplementary Figure 10c), suggesting that using spatial data might alleviate the typically challenging heterogeneity in multifocal PCa for biomarker development [85].”

Figure 4 APC and CEG score variation in multifocal prostate cancer samples. Gene expression data obtained from Strømme et al. 2022 (doi: 10.1038/s41417-022-00444-7) were used to calculate scores for APC (a & b) and CEG (c & d) signatures. Expression data for 6 of the 28 APC signature genes AC069228.1, IGHG3, IGHG4, IGLC2, IGKC, and IGLC7 was missing in the dataset. Shown are the mean scores with standard deviation per patient as bar plot (without, a & c, and with, b & d, benign samples) and the individual scores for each sample as dots color coded according to PCa focus as labeled in Strømme et al. 2022 (F1-F3, Benign).

2. There is an issue in relation to the clinical description of the two groups of cancerous prostates: relapsed and non-relapsed. The definition in the manuscript of relapsed patients is as follows: biochemical recurrence (PSA > 0.1 ng/ml) and/or confirmed metastasis. However, there is a big difference between PSA failure and metastasis particularly at 10 years. The authors need to mention in the main text how many of the five selected cancers are in each group. They also need to give a justification for this selection.

Two of the 5 patients in the aggressive group of the cohort used for spatial transcriptomics data collection had confirmed metastasis (22 month and 90 month post OP) in addition to persistent PCa in one case and biochemical recurrence after 37 month in the other. This was added to Supplementary Table 1 and a corresponding sentence was added to the main text on page 3:

“Of the 8 patients, 5 experienced progression (aggressive patient group) with both persistency (n=4, PSA>0.1ng/ml) and relapse (n=1, PSA>0.2 ng/ml) within 37 months, in addition 2 of these had confirmed metastasis after 90 and 22 months. Three patients remained relapse free, termed non-aggressive, during a follow-up of 10 years although they had similar clinicopathological features at diagnosis (Table 1, Supplementary Table 1).”

The respective section in the methods was changed accordingly:

“In total, tissue samples from 37 patients were included in this study. Among them, ten patients, grouped as non-aggressive PCa, remained relapse-free for >10 years following surgery, while twenty-seven patients, grouped as aggressive PCa, either relapsed (n=16, biochemical recurrence, PSA > 0.2 ng/ml) or were persistent (n=11, PSA > 0.1 ng/ml). In addition, two of the aggressive PCa patients had confirmed metastasis (details in Supplementary Table 1).”

3. There is a critical problem in the overall design of the study. The manuscript was well written and easy to follow. However, there was little statistical power both in the hypothesis generating set (8 patients) and in the validation set (37 patients). There was also an overlap between these two set (the initial 8 patients are include entirely within the 37 validation cohort of patients). This design completely invalidates any statistical meaning irrespective how many samples are taken from each patient.

We thank the reviewer for this critical comment. Following also the suggestion of the other reviewers, we have now added additional validation analysis using data from a total 1588 PCa patients – see response to reviewer 1 comment 1.

4. I found the arguments in the body of the text quite convincing based on the data presented. However, at best, because of the small number of patients, they can only be considered anecdotal, generating a hypothesis that can be tested in a much larger dataset. Unfortunately, the slightly larger dataset presented overlapped with the discovery dataset and in any case was not big enough. To provide a really convincing confirmation a larger dataset of hundreds of patients would be required. In this reviewers experience there are many cases where biomarkers found to be statistically significant in smaller dataset have not been confirmed in much larger dataset (100s of patients). I would encourage the authors to take the time to carry out this confirmation, as the results of their study, if true, are significant. I wonder if a number of publicly available datasets could be used; confirmation in multiple publicly available datasets would make the results very strong. Confirmation on a much larger dataset would be needed if the work is to be published here. In any case the confirmation dataset would need to be entirely independent of the smaller hypothesis generating dataset.

We thank the reviewer for this suggestion. We performed such analysis confirming the APC signature to be predictive of relapse and added these analyses to the manuscript – see reply to reviewer 1 comment 1 for details.

5. The manuscript presents a number of sets of analysis after selecting of the original cancers, not cancer areas and the resulting spots. They first generate a signature called RA that contains many immunologically related genes. A CEG signature was then calculated that at a single spot level (ssCEG) showed enrichment in all classes of spots (cancer, field effect, normal). This was OK but is was not clear why the spots located immediately adjacent to the cancer were designated “field effect”. This implies knowledge of mechanism. The authors should choose something more neutral like “morphologically normal tissue adjacent to cancer”.

We thank the reviewer for this suggestion. We changed the name of the field effect samples to “Normal Adjacent” to avoid such implications.

6. In the next section the authors looked at the RA and CEG in relation to luminal status and citrate secretion. Here I was unconvinced by the rational of fusing cancer and field effect sample; if there were a low number of field effect samples as stated then these should be excluded to make a cleaner

comparison between cancer and normal. Indeed you could argue that the field effect samples should be merged with the normal samples, since they are both non-cancerous. This section needs reworking.

This analysis was based on the classification of the spots, and not the samples. In this context, spots classified as normal appearing glands can be either closer to cancer or further away from cancer. Thus, it makes sense to merge normal appearing spots in cancer samples with normal appearing spots in field samples (now termed ‘Normal Adjacent’), since they are both closer to the cancer, while normal appearing spots in normal samples are far away from the cancer. In that respect we have compared luminal status and citrate secretion between normal spots in cancer and field samples and observed that the pattern is similar (Figure 5).

Figure 5 Citrate secretion gene set activity in different spot groups represented by the average ssGSEA scores derived from spatial transcriptomics data for each group.

Moreover, since there was also only one Recurrent Field sample (now termed ‘aggressive normal adjacent’), we found it better to merge the normal appearing field spots with the normal appearing cancer spots rather than present results from only one sample

7. The authors next present evidence that RA and GEG signatures are associated with Club-like cells and immune cell enrichment. This section provides some very interesting anecdotal observations that need confirming in much larger independent datasets. In the following section the authors headline that the RA and CEG signatures arise from Club-like cells. The arguments were a bit tenuous and not as convincing as other aspects of the paper. Based on the correlation evidence presented the authors could conclude that an origin in club-like cells is one of the options so the work “likely” in the title is overstating the case. This section would need modifying so that conclusions from the data are not overstated. Other sections would need modifying also. Correlations need to be presented in a more “lay friendly” way. “($\rho \geq 0.25$)” does not mean much to most people; what is the statistical significance? Please rework this.

We agree with the reviewer that a larger set of spatially resolved gene expression data or possibly carefully designed cell-biology studies would increase confidence in our findings. However, the spatial gene expression data we collect stems from 8 different individuals and in total from 19854 sampling spots sampling approximately 20 cells per spot. Thereby outperforming typical genetic and cell type variability of commonly used and widely accepted cell biology studies based on a few selected cell lines.

We would like to emphasize that we do not claim that the APC and CEG arise from Club-like cells, but we present that some of the genes are mainly correlated with Club-like cells. Of those, LCN2, LTF, MUC5B, CXCL1, CXCL6, and CXCL17 are known to be enriched in effector Club cells of the lung (see Discussion, page 19) and notably also the other chemokines of the signatures. We changed the headline of this section and the conclusion drawn in the last sentence to better reflect this observation:

“APC and CEG activity correlate with presence of chemokine-secreting Club-like cells”

“Considering that Club-like cells are known to secrete chemokines [14], these correlations suggest Club-like cells as one likely source for the chemokines of our signatures.”

The requested “lay friendly” presentation of the spearman correlation coefficients ρ are in our opinion given in form of the text before the values in parentheses: “...we found clear positive correlation ($\rho \geq 0.25$) for the genes...” and in the heatmaps presented in the Supplementary Figures 6 and 7. The values in parentheses given for reference. The absolute values should not be overinterpreted and the focus should rather be on the direction (negative or positive correlation) and relative differences such as highest observed correlated in our data. Calculation and reporting of p-values is, in our opinion, meaningless as it would simply allow rejection of the null hypothesis that the respective correlation coefficient is not 0. Due to the high number of data points (~20000) even the smallest non-zero correlation coefficients will result in p-values < 0.001 . We edited the whole section to better reflect the focus on relative comparison and are confident to have found a sufficient compromise.

8. The next section presents a case that Atypical Chemokine Receptor 1 (ACKR1) is the dominant receptor for the CEG signature. This conclusion seemed to be based on the observation that only expression of CXCR4 and ACKR1 was detected in the ST data. This section was more convincing but again there needs to be a more lay friendly presentation of statistically significant correlations. The remaining sections were also convincing.

We reworded the last part of this section presenting the correlations as detailed in our answer to the previous comment on page 11 in the paragraph “Atypical Chemokine Receptor 1 (ACKR1) is the dominant receptor for CEG-signature CXC-chemokines in the aggressive patient group”:

“The strongest and single distinct positive correlation of ACKR1 was observed with endothelial cells ($\rho = 0.29$) and the strongest negative correlation with luminal epithelial cells ($\rho = -0.12$). In contrast, DPP4 was negatively ($\rho = -0.20$) correlated with endothelial cells and strongest positive correlation was observed with luminal epithelial cells ($\rho = 0.53$). CXCR4 displayed highest positive correlation with cells of the myeloid ($\rho = 0.37$) and lymphoid ($\rho = 0.28$) lineages and, like ACKR1, also strongest negative correlation with luminal epithelial cells ($\rho = -0.26$).”

9. Discussions were a bit long winded and I recommend shortening by 50% or so for a final manuscript. There seemed to be a lot of unnecessary detail. Just keep to key discussion points.

We choose to discuss the individual genes of both signatures in detail adding substantially to the here criticized volume of the discussion. In our opinion, such a detailed discussion adds essential value to the biology and meaning behind these biomarkers and are required to untangle the complexity of the tumour microenvironment. It appears that reviewer 4 (comment 2) asks to further broaden the discussion. We are open to editorial input on this matter.

Overall this was an interesting paper that contained potentially significant novel observations. However many observations were anecdotal only. The study design had critical flaws that need addressing before the paper could be accepted.

We hope that we could satisfactorily answer all questions of reviewer #2 and specifically convince the reviewer of the validity of our findings with the newly added results from validation of our signatures in 1588 patients.

Reviewer #3 (Remarks to the Author):

In the manuscript titled "Deep Phenotyping of the Prostate Tumor Microenvironment Reveals Molecular Stratifiers of Relapse Linked to Inflammatory Chemokine Expression and Aberrant Metabolism" Sebastian Krossa et al. apply spatially resolved profiling methods to determine molecular features of prostate cancer foci with inferred aggressive behavior. Broadly, risk stratification for localized prostate cancer represents a major unmet medical need. While numerous expression signatures have been introduced in recent years aimed at addressing these needs, this study takes a different approach by applying spatial transcriptomics and metabolomics profiling to primary tumor specimens. In a relatively limited set of samples, the authors find that gene expression signatures associated with inflammation and a previously defined epithelial cell type ("club-like cells") show an association with disease behavior. While this manuscript represents an interesting descriptive study of a small number of specimens, it lacks focus and clear clinical relevance. There are several other points to consider:

1. The number of samples in the discovery set is very low, and extensive external validation in datasets with long-term follow-up is missing.

We thank the reviewer for this critical comment which was also highlighted by the other reviewers. We have performed external validations of both signatures and added the results to the manuscript – see reply to reviewer 1 comment 1 for further details.

2. The identification of "benign" areas with cancer-associated changes is challenging to contextualize given the relatively low resolution of the method used. In situ validation and documentation of this finding through detailed histologic analysis would be necessary.

All histo-pathological evaluation and resulting annotations of regions and consequently ST spots has been performed using microscopic images of H&E stained tissue as described in the respective methods sections. In our opinion, the reported resolution is sufficient to justify the chosen region/area based analysis (such as cancer with grade, non-cancer gland, stroma). We agree that the resolution is not at the same level of for example typical microscopic images after IHC. Nevertheless, the high data redundancy and much higher plexity obtained by the amount of datapoints collected using ST allows mathematical/statistical approaches to deconvolve cell-type compositions (Andersson, A. *et al.* Single-cell and spatial transcriptomics enables probabilistic inference of cell type topography. *Commun Biol* **3**, 565 (2020). doi: 10.1038/s42003-020-01247-y) or super-sample this data (Zhang D et al. Inferring super-resolution tissue architecture by integrating spatial transcriptomics with histology. *Nat Biotechnol.* 2024 Sep;42(9):1372-1377. doi: 10.1038/s41587-023-02019-9.). We chose to utilize only the deconvolution in this study as we found the results to be significantly more useful and approachable.

3. The general theme that inflammation in the prostate is associated with carcinogenesis and adverse clinical features is, in itself, not novel. There is substantial literature on putative inflammatory precursors, such as PIA, that show cancer-associated changes.

We agree with the reviewer that the link between cancer and inflammation is not novel. Nevertheless, the results of deep phenotyping approach point at specific cell types composing the tumor microenvironment and therefore our opinion is that our results contribute to deepen the knowledge and understanding of the complex interplay between cancer cells and inflammation in the spatial context / with multiple multi-omics layers which is a novel approach. In particular we could validate the interplay between club cells and inflammation which we have previously reported also in Kiviaho et al., Single cell and spatial transcriptomics highlight the interaction of club-like cells with immunosuppressive myeloid cells in prostate cancer. Nat Commun. 2024 Nov 16;15(1):9949. doi: [10.1038/s41467-024-54364-1](https://doi.org/10.1038/s41467-024-54364-1).

Reviewer #4 (Remarks to the Author):

This study identified two immune-associated signatures, characterized by increased chemokine production and dysregulated citrate metabolism, that are selectively linked to aggressive prostate cancer via integration of spatial multi-omics (transcriptomics and metabolomics). Notably, the authors reported that this occurred even in non-cancerous glands of patients who relapsed after surgery, and further speculated that immune infiltration and inflammation in the tumor microenvironment may be a driver of progressive disease. The manuscript also suggests the potential utility of chemokines as biomarkers of relapse, with linkages to metabolism through modulation of citrate.

Positive aspects of the study include the use of multiple sampling sites in each patient tumor to incorporate intratumoral heterogeneity and the microenvironment, and that the authors excluded prostatitis as a potential, though rare, confounding factor in the gene signatures. However, the study design fundamentally compromises interpretation of the study's findings, particularly concerning patient selection in a very small cohort. The reported outcomes therefore currently do not adequately support the authors' conclusions. Additionally, the authors need to address several major concerns, which are detailed in the review below.

Comments:

1. In Supplementary Table 1, column 3, 'Time to relapse (months)', could the authors clarify the '0' values for a subset of relapse patients? Currently, it appears that '0' means patients who 'relapsed' actually were cases of non-curative surgery where PSA levels did not decrease to undetectable levels and, therefore, depending on the definition, did not represent a true relapse – these patients are already known to have poorer outcomes, likely due to pre-existing metastatic spread or locoregional disease. Notably, 4 out of 5 patient samples used for spatial transcriptomics had the '0' time to relapse, which is not an ideal set of samples for this comparative study, and 11 out of the 27 broader cohort. The tumor grades across patients were imbalanced and particularly higher for patients who 'relapsed', and 4 of the relapsed patients had pre-operative PSA of over 30. Given the constraints of cost in undertaking this type of multi-omic approach, which necessitates smaller sample numbers, it is even more important to control for these clinical factors as much as possible and include only patients who experienced an

apparently curative surgery. I believe the authors characterize a completely different biology here, based on the stage of disease progression rather than a signature of relapse.

The intention was to characterize the biology of aggressive PCa by selecting the most aggressive PCa patients from our biobank. We agree with the reviewer, that using the label relapse was misleading. We renamed the groups into non-aggressive and aggressive and changed text and figures of the whole manuscript accordingly. Further, the newly added validation with accurately defined outcomes (biochemical recurrence and metastasis) of the APC (former RA) signature proves that our approach was successful and we conclude that we provide important and valid insights into the characteristics of the TME of aggressive PCa. Please also see details of the validation by reading answer to comment 1 from reviewer 1.

2. In the discussion, the authors briefly described the function of each gene derived from the immune-associated signatures and their known role(s) in cancer, but do not elaborate on the biology that is uncovered. Is there a distinct population of immune cells driving this phenotype? What are the drivers of this response in non-cancerous cells? Could the authors also perform cell-type deconvolution on the bulk transcriptomics data to confirm this observation in the spatial data?

As discussed in the manuscript we believe the Club-like cells, although not belonging to any “classical” immune cell population, to be the main “driver” of the observed phenotype. To the best of our knowledge, no model exists that could allow meaningful deconvolution of bulk RNA-seq data for resolving the interplay between immune cells and club-like cells. However, we recently reported that club-like cells specifically interplay with myeloid immune suppressive cells (Kiviahio et al. 2024, doi: 10.1038/s41467-024-54364-1) and in our correlation analysis on data from 12 public data sets we do observe consistent high correlation with immune cells of myeloid lineage (Figure 7), supporting the conclusion that Club-like cells are the most likely driver of this phenotype.

3. Concerning the discrepancy observed in the comparison between spatial transcriptomics and bulk transcriptomics data, how representative are the patient tissues used for bulk transcriptomics compared to the patient tissues used for spatial omics? Based on the table, it looks like tissues used for bulk transcriptomics had higher grades of tumors, but this is not accounted for.

We thank the reviewer for this comment which allowed us to further clarify this point. We would like to emphasize that the bulk data used includes also bulk data obtained from the same cores from which tissue for spatial transcriptomics originated. For those 8 patients and 32 cores we see higher similarity in signature scores (Figure 6). We do not follow on the stated discrepancy; the signal was just not as prominent in bulk as it was in spatial data. This is not surprising due to the very different nature of these data as the bulk data originates from a much larger tissue volume and “spatial” signals are more prone to get averaged out and “suppressed” in this case. We believe that this is one of the main reasons why the CEG signature, extracted from the microenvironment, loses its power in bulk data while the for the APC signature, designed to be less dependent on spatial data, gets just less intense in bulk samples containing no cancerous tissue. Considering the commented grade composition of bulk and spatial data we like to clarify that the grades given in Table 1 are the patient level ISUP grades. Each core was graded again based on individual histo-pathology evaluation and spatial data had in addition a much more detailed spot level grading used for Figure 2 b and c and is not included in Table 1. Based on the numbers given in Table 1, on the patient level, the aggressive bulk cohort consisted of 26% ISUP2, 37% ISUP3, 15% ISUP4, 22% ISUP5 and the non-aggressive of 80% ISUP2 and 20% ISUP3 while the “spatial” subset consisted of 20% ISUP2, 20% ISUP3, 20% ISUP4, 40% ISUP5 (aggressive) and 67%

ISUP2 and 33% ISUP3 (non-aggressive). Thus, showing a slightly higher amount of low-grade patients in the full “bulk” cohort.

Figure 6 APC and CEG signature scores in spatial transcriptomics (ST) and bulk RNA sequencing data from corresponding cores.

4. Could the authors expand and validate the observed immune signatures in a spatial context in an independent cohort of patient tissues, possibly addressing the issues of clinical relapse discussed in point 1 above?

We are unfortunately not aware of any published large cohort with spatially resolved data and follow up associated for performing this type of validation. However, we used bulk transcriptomics data to validate our signatures in a significantly larger set of patients. See reply to Reviewer 1 comment 1 for further details.

5. In Figure 2A, the localization of the signature signals does not appear to very clearly match their pathological features (e.g., NCG). It would be helpful to include H&E images for each tissue in this figure to show how well the ST deconvolution reflects the histopathology and vice versa.

We thank the reviewer for the suggestion. However, in our opinion the addition of several images of H&E stained tissue to the main Figure 2A will not achieve the desired intent of clarification. We however present here an example for the reviewer to evaluate how the scores of the signatures in the spots match the H&E staining (Figure 7 & 8).

Considering that each ST spot consist of on average 20 cells (see Supplementary Figure 2 a) and cover an area of ~2500 μm^2 results on an individual spot level will always be relatively blurry. We rather see value correlating the dominant histopathological characteristic, derived from HE images, of each spot with the estimated cell-type data composition (as shown in Supplementary Figure 6) to prove validity of this approach. In a similar way, Figure 2b together with Suppl. Fig. 4 b & d demonstrate that the CEG signal originates mainly from NCG spots but is not exclusive to it.

Figure 7 H&E stained section of sample P30 - core C2 overlaid with spatial transcriptomics spots color-coded according to CEG score (red - white - blue corresponding positive - zero - negative CEG score)

Figure 8 Magnified area of section of sample P30 - core C2 shown in Figure 6 containing glandular and stromal tissue.

6. Were the authors able to determine the association of signatures with progression-free survival in publicly available bulk datasets? Looking at the datasets used in Supplementary Table 3, some of the datasets (ie. Taylor and Sboner) have outcome data. The authors could try to compare their signatures in more aggressive prostate cancer samples (ie. metastatic samples) rather than just normal vs cancer.

We added results from validation of our signatures in several cohorts to the manuscript. See reply to Reviewer 1 comment 1 for further details.

7. For the metabolomics data, the data are not clearly described in the text and the biological significance is not clear – was the metabolism perturbed in a specific cell-type? Besides the correlation between gene and metabolites of citrate metabolism, were there other noteworthy correlations at both the transcriptional and metabolic level?

We thank the reviewer for the opportunity to further clarify this point. Considering the well-established knowledge that a drastic reduction in citrate levels is characteristic for prostate cancer we were mainly interested in looking into the correlation of citrate level spot level and choose to focus on citrate-related

changes in regions showing high activity of our signatures. We have now rephrased the text in multiple point to further clarify the intent of evaluate the metabolic changes associated with areas/spots (not cells) with high signature activities. See also response to reviewer 2 comment 7.

8. There are known ambiguities in identifying metabolites using untargeted mass spectrometry imaging—there is no mention of the m/z or ppm/mDa range in the manuscript. In their methods section, “metabolites were identified by MS/MS and accurate masses from our previous study using the same methods for equivalent PCa samples”. Was MS/MS performed on these specific patient samples or inferred from the previous study? When making significant claims about correlations between analytes, and especially in this study their relationship to disease progression, MS/MS should ideally be on these exact samples or if not possible, this should be discussed as a study limitation.

We realize the previous description was somewhat misleading. In our previous studies as well as this study, prostate tissue sections were covered by NEDC and DHB matrix and analyzed with a Rapiflex MALDI-TOF MSI instrument. For identification of a selection of the masses we used MALDI-Orbitrap to perform non-imaging MS/MS and to acquire high-mass resolution spectra from a few of the matrix-covered tissue sections. Accurate mass was defined as below 3 ppm. To the best of our knowledge, it is accepted practice in MSI to identify masses on a small subset of the dataset and corresponds to a level of confidence of 2/3 according to Baquer et al. 2022. Since the previous studies used PCa tissue sections that had been processed the same way as in this study, the mass identifications are valid.

A table of identified masses and the m/z ranges (+/- 400 ppm) used for ion image generation is added to the supplementary (see Supplementary File 5).

We have added additional clarifications to the methods text.

“A selection of masses was identified as metabolites by MS/MS and accurate masses (< 3 ppm) from our previous study using MALDI-Orbitrap [97], while Zinc was detected as an adduct with chloride (ZnCl₃⁻) and was previously identified by its isotopic pattern [22]. All previous identification work was performed on fresh frozen PCa sections that had been collected and processed the same way as in this study and were covered by the same matrix. The identifications are therefore valid for this study and are on confidence level 2-3 as suggested by Baquer et al. 2022 [98].“

9. Notably, the authors did not present any spatial metabolomics data in their main figures or supplementary figures, which is an important omission. In Figure 5, including only box plots without any images or spectra is not showing the data, and while alterations in metabolite abundance are claimed based on these box plots, this is unconvincing and no statistical proof is provided. Representative m/z images of each metabolite should at a minimum be shown – boxplots are informative but can be misleading. For example, comparing two ROIs between tissue sections can show a significant increase in a mass abundance that, when compared to the image, can reveal a background or matrix ion that, by chance, has higher intensities in one of the ROIs compared to the other.

Data presented in Figure 5 includes data from spatial metabolomics in form of box-and-whisker plots as indicated. We added files with MSI images of all metabolites obtained after applying our integration tool MIIT (doi: 10.1093/gigascience/giaf035) to the supplement, “raw”, unregistered ion images before applying MIIT, and average mass spectra for visualizing the individual m/z ranges chosen:

- Supplementary File 6:
 - Supplementary_File_6_MSI_ion_images_registered_to_ST_using_MIIT.pdf

- Supplementary File 7:
 - Supplementary_File_7_MSI_mean_spectra.pdf
- Supplementary File 8:
 - Supplementary_File_8_MSI_unregistered_ion_images.pdf

Any statistical test performed on large sample sizes typically obtained with spatial omics techniques will result in very low p-values for even the smallest differences. In our opinion, this is misleading, and we chose to abstain from using tests and rather show measured distributions to visualize effect sizes. We opted for boxplots because they are relatively easy to read even though they are not allowing visualizations of bimodal distributions.

Regarding the reviewer's comment on potential contributions on background signals, we agree that we need to provide additional data to support our data interpretation. Ion images can help here but cannot necessarily easily reveal contribution of a matrix or background ions on their own. An ion image is typically generated from the MS data by displaying the highest intensity in a reasonably selected mass range for each individual "pixel"/mass spectrum. We agree that a too broad selection of such mass ranges can lead to false ion images by including background/matrix peaks that have a higher intensity than the actual "ion of interest" in this range in some pixels. In very extreme cases, such errors might be directly apparent in the ion images but inspecting the mass spectra for such potential overlaps and for example compare them to background spectra collected "off-tissue" in combination with inspection of isotopic patterns is likely a better strategy. Nevertheless, to obtain a comparable high confidence, as in for example LC-MS/MS analysis, of the identity of each peak of each individual pixel would require at least MS2 data for all peaks in all pixels. As already mentioned in the response to the previous comment, this is not yet feasible for MSI. We hope that the added data sufficiently demonstrates and convinces the reviewer that we obtained an acceptable level of confidence of our identifications and derived metabolite intensities per ST spot and that conclusions drawn are sufficiently supported by our data.

Minor comments:

1. In Supplementary Figure 4, high RA signature activity was mostly detected in the stromal regions of the tissue (based on histopathology). However, the authors claim enrichment of club-like epithelial cells associated with the RA signature.

Both aspects are not mutually exclusive. The RA signature (now renamed APC) was designed to be as independent of tissue class as possible. Thus, it is possible and reasonable to think that the APC signature contains both aspects of stroma associated with prostate cancer as well as it is associated with club-like cell enrichment. Further, we like to point out that we observed that much of that association of APC activity with stroma stems from "inflamed", immune cell infiltrated stroma (see Figure 2 b) – a distinction we dropped for Supplementary Figure 4. The 5 immune globulins in the APC signature are the most likely candidates to account for this association.

2. In Supplementary Table 1, the patients with 'control' status had months indicated for 'time to relapse'. As these patients did not by definition relapse, for clarity this column should indicate N/A with the number of months actually reflecting the follow up time rather than time to relapse.

We agree with the reviewer. The 'time to relapse'-column of the Supplementary Table 1 has been updated by replacing the numbers to the symbol "∞" for the non-aggressive patients.

3. For the signature generation, was there a reason for selecting only the top 5 differentially expressed genes? Did the authors try to evaluate the robustness of the signatures?

Selecting the top 5 differentially expressed genes was an arbitrarily chosen cutoff to limit the number of input genes for signature generation. The new and larger scale cohort validation analyses included for this revision shows that the output was robust and the signatures were stable also in bulk gene expression data.

Reviewer #5 (Remarks to the Author):

We thank the reviewer for participating in the review process.